# Treatment of monogenic and digenic dominant genetic hearing loss by CRISPR-Cas9 ribonucleoprotein delivery in vivo

Yong Tao[1,2,3,15], Veronica Lamas[1,2,13,15], Wan Du[1,2,15], Wenliang Zhu[1,2,15], Yiran Li[1,2,14], Madelynn N. Whittaker [4,5], John A. Zuris[6,7,8], David B. Thompson[6,7,8], Arun Prabhu Rameshbabu [1,2], Yilai Shu [1,2,9], Xue Gao [6,7,8], Johnny H. Hu[6,7,8], Charles Pei[2], Wei-Jia Kong [10], Xuezhong Liu[11], Hao Wu[3], Benjamin P. Kleinstiver [4,5,12], David R. Liu[6,7,8] ✉ & Zheng-Yi Chen [1,2] ✉

Mutations in *Atp2b2*, an outer hair cell gene, cause dominant hearing loss in humans. Using a mouse model *Atp2b2^{Obl/+}*, with a dominant hearing loss mutation (Oblivion), we show that liposome-mediated in vivo delivery of CRISPR-Cas9 ribonucleoprotein complexes leads to specific editing of the *Obl* allele. Large deletions encompassing the *Obl* locus and indels were identified as the result of editing. In vivo genome editing promotes outer hair cell survival and restores their function, leading to hearing recovery. We further show that in a double-dominant mutant mouse model, in which the *Tmc1* Beethoven mutation and the *Atp2b2* Oblivion mutation cause digenic genetic hearing loss, Cas9/sgRNA delivery targeting both mutations leads to partial hearing recovery. These findings suggest that liposome-RNP delivery can be used as a strategy to recover hearing with dominant mutations in OHC genes and with digenic mutations in the auditory hair cells, potentially expanding therapeutics of gene editing to treat hearing loss.

Deafness affects about 466 million people, including 34 million children, constituting 6% of the global population (http://www.who.int/en/). The most common type is sensorineural hearing loss (SNHL) of which pathogenesis occurs in the inner ear, the sensory organ or the auditory nerve. Hereditary types of hearing loss account for more than 50% of all congenital sensorineural hearing loss cases and are caused by genetic mutations[1]. To date, over 120 genetic loci have been linked to hereditary hearing loss (http://hereditaryhearingloss.org/), but

[1]Department of Otolaryngology-Head and Neck Surgery, Graduate Program in Speech and Hearing Bioscience and Technology and Program in Neuroscience, Harvard Medical School, Boston, MA 02115, USA. [2]Eaton-Peabody Laboratory, Massachusetts Eye and Ear Infirmary, 243 Charles St., Boston, MA 02114, USA. [3]Department of Otolaryngology-Head and Neck Surgery, Shanghai Ninth People's Hospital, Shanghai Jiao Tong University School of Medicine, Shanghai 200011, China. [4]Center for Genomic Medicine, Massachusetts General Hospital, Boston, MA 02114, USA. [5]Department of Pathology, Massachusetts General hospital, Boston, MA 02114, USA. [6]Merkin Institute of Transformative Technologies in Healthcare, Broad Institute of MIT and Harvard, Cambridge, MA, USA. [7]Department of Chemistry and Chemical Biology, Harvard University, Cambridge, MA, USA. [8]Howard Hughes Medical Institute, Harvard University, Cambridge, MA, USA. [9]ENT Institute and Department of Otorhinolaryngology, Eye & ENT Hospital, State Key Laboratory of Medical Neurobiology and MOE Frontiers Center for Brain Science, NHC Key Laboratory of Hearing Medicine, Institutes of Biomedical Sciences, Fudan University, Shanghai 200031, China. [10]Department of Otorhinolaryngology, Union Hospital, Tongji Medical College, Huazhong University of Science and Technology, Wuhan, Hubei, China. [11]Department of Otolaryngology, University of Miami Miller School of Medicine, Miami, FL, USA. [12]Department of Pathology, Harvard Medical School, Boston, MA 02115, USA. [13]Present address: Salk Institute for Biological Studies, La Jolla, CA 92037, USA. [14]Present address: C.S. Mott Children's Hospital, Ann Harbor, MI, USA. [15]These authors contributed equally: Yong Tao, Veronica Lamas, Wan Du, Wenliang Zhu. ✉e-mail: dliu@broadinstitute.org; Zheng-Yi_Chen@meei.harvard.edu

there are currently no biological treatments for any form of hereditary deafness[2,3]. In the last two decades, the use of gene therapy strategies as treatment for inner ear dysfunction has emerged as a powerful therapeutic approach in deaf mouse models. Gene replacement using adeno-associated viral (AAV) vectors, gene silencing using either antisense oligonucleotides or AAV delivered microRNAs are strategies that have shown promising results in improving hearing in mouse models of deafness[4–8]. However, these approaches have limitations including the size of genes to be inserted into an AAV, inefficient targeting dominant mutations, and potential long term safety concerns of viral vectors.

The advent of genome editing has made it possible to treat genetic diseases by the modifications at genomic DNA level to disrupt or correct the mutations. Furthermore, the recent discovery and development of CRISPR-associated protein 9 (Cas9)-based genome editing has facilitated the effective targeted gene disruption or repair of virtually any sequence of interest in the genome, and has been used to treat hereditary diseases in mouse models[9–13]. We and others have applied the CRISPR-based gene editing agents to recover hearing in mouse models of human genetic deafness. By lipid and AAV mediated delivery of *Streptococcus pyogenes* Cas9 (SpCas9) ribonucleoprotein (RNP) complexes, we and others have improved hearing in a mouse model of dominant hearing loss of hair cell origin, the Beethoven (*Bth*) mouse, by targeting the mutant allele of the transmembrane channel-like 1 (Tmc1)[14,15]. Recently, the new generation of base editors were used to repair a recessive mutation on the *Tmc1* gene, which transiently improved hearing in the Baringo mice[16]. The editing treatment for hearing loss, however, has been limited to inner hair cells (IHC) due to the primary role of *Tmc1*. The auditory function requires functional inner and outer hair cells (OHC). Besides, multiple types of genetic hearing loss are associated with mutations in OHCs[17]. In our editing treatment of the Beethoven mutant mice for hearing recover, we detected deterioration in OHC function shown by distortion product of acoustic emissions (DPOAE) in the treated inner ears[14], raising the question if our liposome-mediated delivery and editing is suitable to target OHC mutations to recover hearing. Thus, it is necessary to address the limited efficiency of gene editing therapies in targeting and functionally rescuing the OHC.

To evaluate liposome delivery of gene editing complex targeting the OHCs, we selected a dominant deafness mouse model (*Atp2b2^Obl/+^* mouse), which carries a mutation Oblivion (*Obl*) in the second isoform of the plasma membrane $Ca^{2+}$-ATPase (*Atp2b2*) gene that impairs the calcium pumping ability of the PMCA2 protein[18]. The PMCA2 calcium pump is produced at high level in the stereocilia bundle of the OHC[19]. Mutations of PMCA2 $Ca^{2+}$ pump of the stereocilia cause deafness and loss of balance in both mice and humans[18,20,21]. *Obl* heterozygous mice show progressive hearing loss starting at P20 and progressive degeneration of the sensory hair cells. *Obl* homozygous mutants are completely deaf from birth and show a severe degeneration of the organ of Corti. In this work, we apply our previously described CRISPR-Cas9 lipid-mediated delivery paradigm to the Oblivion mouse model to treat hearing loss of the OHC origin. We first selected and evaluated the ability and specificity of different sgRNAs in disrupting the *Obl* mutation in vitro. Then, we performed lipid-mediated delivery of Cas9 and sgRNA ribonucleoprotein complexes into the inner ear of postnatal heterozygous *Atp2b2^Obl/+^* mouse to recover hearing.

Although most cases of genetic deafness result from mutations at a single locus, an increasing number of examples are being recognized in which mutations at two loci are involved. For example, digenic interactions are known to be an important cause of deafness in individuals who carry mutations at the Connexin 26 and 30 loci[22], or Connexin 26 and 31 loci[23]. Pathogenic digenic inheritance has also been seen in Pendred[24] and Usher syndromes[25], and an *Atp2b2* mutation was found in a deafness family with a homozygous *Cdh23* mutation[20]. For these cases, editing therapy that targets both mutations will be necessary for hearing recovery. Here, we performed editing to target two mutations simultaneously, the Oblivion mutation in the *Atp2b2* gene and the Beethoven mutation in the *Tmc1* gene, in a deaf mouse model that harbors both mutations, which led to partial hearing recovery. Our results strongly support the feasibility of liposome-mediated delivery of editing RNP to target dominant mutations of auditory hair cells in hearing recovery.

## Results

### Screening sgRNAs in vitro

To develop a genome-editing strategy capable of disrupting the mouse *Obl* mutant allele, we began by searching for protospacer sequences at the target site. We identified three sgRNAs that target *Atp2b2* at the sites that include the *Obl* (c.C2630T, p.S877F) mutation and a nearby NGG protospacer adjacent motif (PAM) sequence required by SpCas9. The three candidate sgRNAs (*Atp2b2*-mut1, *Atp2b2*-mut2, and *Atp2b2*-mut3) place the *Obl* mutation at the position of 18, 12, and 20, respectively, of the protospacer, counting the PAM as positions 21-23 (Fig. 1a). We evaluated the ability of these sgRNAs when complexed with Cas9 to cleave either the wild type (WT) or the *Obl* allele in vitro. All three sgRNAs cleaved the *Obl* allele, with *Atp2b2*-mut1 exhibiting the greatest selectivity for the *Obl* over the wild-type *Atp2b2* gene (Fig. 1b, c). To better evaluate the specificity of genomic editing on the *Obl* allele, we generated a cell line, Obl-OC1, by inserting a mouse genomic DNA fragment harboring the *Obl* mutation into the WT mouse inner ear cell line, HEI-OC1[26]. We isolated the genomic DNA from cells transfected with Cas9 protein complexed with each of the three sgRNAs by nucleofection and performed T7E1 assay and NGS analysis to evaluate cleavage and the indel frequency at the target sites (Supplementary Fig. 1a, b). The gRNAs Atp2b2-mut2 and Atp2b2-mut3 resulted in cleavage in WT and *Obl* Atp2b2 DNA (Supplementary Fig. 1a) from the T7E1 assay. The gRNA Atp2b2-mut1 produced the cleavage products only in the Obl-OC1 cells (Supplementary Fig. 1a), an indication of high selectivity at the *Obl* allele by Atp2b2-mut1 gRNA. NGS analysis showed a high rate of indels on the *Obl* allele in the Obl-OC1 cells with 34.8% from Cas9:Atp2b2-mut1, 79.5% from Cas9:Atp2b2-mut2, and 66.4% from Cas9:Atp2b2-mut3 treatment (Supplementary Fig. 1b, c). The indel analysis showed that Cas9:Atp2b2-mut1 exhibited the highest specificity on the *Obl* site with no detectable editing events in the WT Atp2b2 locus from HEI-OC1 cells (Supplementary Fig. 1b, c). In contrast, Atp2b2-mut2 and Atp2b2-mut3 edited the WT HEI-OC1 cells.

To further evaluate the allele specificity of genomic modification in mouse primary cells, we treated mouse primary fibroblasts with Cas9:*Atp2b2*-mut1 by plasmid DNA nucleofection. After nucleofection, we measured the insertion and deletion mutation (indel) efficiency at the on-target by next-generation sequencing (NGS) and detected indels (1.24–2.67%) in both *Atp2b2^Obl/+^* and *Atp2b2^Obl/Obl^* mouse primary fibroblasts (Fig. 1d, e). In contrast, no indels were detected in the wild type (WT) control mouse primary fibroblasts (Fig. 1d, e). The common small DNA modifications were 4 bp deletion and 1 bp deletion around the predicted Cas9 cleavage site (Fig. 1e). The results support that *Atp2b2*-mut1 mediated the most efficient discriminatory editing on the *Obl* allele (Fig. 1e) and thus was selected in the *Obl* mouse model study.

### Off-target analysis

The design of genome editing-based therapies requires the assessment of potential off-target effects to determine the specificity of editing, as the outcome of editing therapy relies on efficient on-target editing in order to be efficacious and the reduction in off-target effects to minimize potential safety concerns. As CRISPR-Cas agents may modify both on-target and off-target loci[27,28], we studied potential off-target loci that could be cleaved by Cas9:*Atp2b2*-mut1 by computational prediction[29] and the GUIDE-seq method[27]. Computational prediction identified five potential off-target sites (Supplementary Fig. 2a)

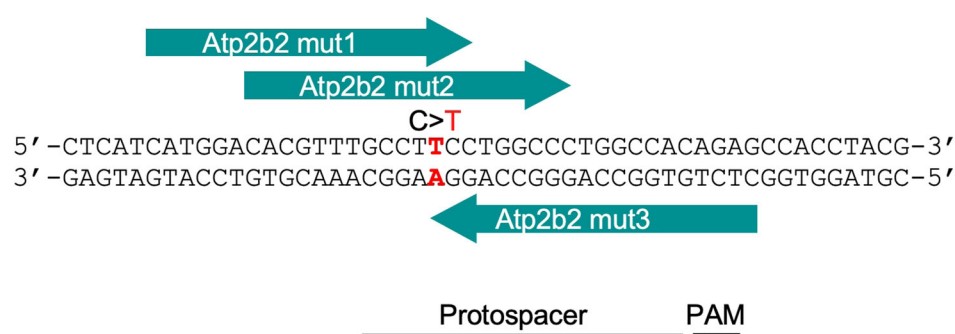

**a   Targeted region of the *Atp2b2^Obl* allele**

(figure panels a–e)

**Fig. 1 | Design of a genome-editing strategy to disrupt the *Obl* mutant allele.**
**a** SpCas9 sgRNAs were designed to target the mutant *Atp2b2 Obl* allele, in which C2765 is changed to T (red). The protospacer (green arrows) of each *Obl*-targeting sgRNA contains a complementary A or T (red) that pairs with the mutation in the *Obl* allele, but that forms a mismatch with wild-type *Atp2b2* allele. **b** In vitro Cas9:sgRNA-mediated *Atp2b2* DNA cleavage. 50 nM of a 956-bp DNA fragments of *Atp2b2* or the *Atp2b2^Obl/+* was incubated with 300 nM of each of the three Cas9:sgRNA for 15 min at 37 °C. Expected cleavage products of 553-bp and 403-bp were detected in all samples. M = 100 bp ladder. This was repeated independently for 3 times with similar results. **c** Quantification of DNA cleavage in (**b**) by densitometry using imageJ. **d** Editing shown by the indel percentages in three unsorted adult mouse primary fibroblast cells (WT, *Atp2b2^Obl/+* and *Atp2b2^Obl/Obl*) after nucleofection of the RNP complex of Cas9:*Atp2b2*-mut1 gRNA. Indels were only detected in the cells with the *Obl* allele. n = 3 biologically independent experiments. Values and error bars are mean ± SD. Source data are provided as a Source data file. **e** NGS reads of indel from (**d**). The *Obl* mutation was highlighted in yellow. The red arrow indicates the cutting site. The list of indels is not comprehensive, only the top representative indels are shown.

containing up to three mismatches in the protospacer region of *Atp2b2*-mut1 sgRNA by computational prediction using the CRISPR Design Tool[29]. None of these loci are associated with hearing function (Supplementary Fig. 2b). We measured the indel frequency at each off-target site by NGS analysis in Cas9:*Atp2b2*-mut1 treated *Obl*-OC1 cells following nucleofection of RNP or plasmid DNA. Plasmid nucleofection resulted in 0.69% indels in the WT *Atp2b2* locus and 0.16% indels in Off-T4 site (Supplementary Fig. 2c). In contrast, RNP treatment exhibited no detectable editing events in either WT allele or other off-target sites (Supplementary Fig. 2c), consistent with the editing in the OC1 cell lines (Supplementary Fig. 1c) and our previous findings that RNP delivery greatly reduces editing at off-target sites[9,14]. We further performed GUIDE-seq to unbiasedly analyze cleaved off-target loci in *Obl*-OC1 cells nucleofected with Cas9:*Atp2b2*-mut1 RNPs. Only the *Atp2b2* locus was identified by GUIDE-seq (Supplementary Fig. 2d). Consistent with NGS analysis of computationally predicted sites, no off-target loci were observed by GUIDE-seq method in *Obl*-OC1 cells nucleofected with RNP. As control we performed QUIDE-seq on the gene *Vbp1* after nucleofection with the Cas9:*Vbp1*-gRNA RNP in *Ob1-OC1* cells. QUIDE-seq analysis identified indels at multiple *Vbp1* off-target sites (Supplementary Fig. 2e). Together, these results support that delivery of Cas9:*Atp2b2*-mut1 RNP complexes into *Obl*-OC1 cells leads to no detectable off-target modification, and that any potential phenotype affecting hearing due to editing are unlikely to arise from off-target modifications.

## In vivo editing with Cas9:*Atp2b2*-mut1 results in indels and large deletions

Efficient editing in hair cells in vivo is likely one of the most important factors determining the extent and duration of hearing recovery in the *Atp2b2*^*Obl/+*^ mice. Hair cells constitute a small percentage of all cochlear cells, which makes it challenging to measure editing efficiency in hair cells when DNA from whole cochleae is used for analysis. In order to more precisely characterize editing at the *Obl* locus in hair cells in vivo, we injected Cas9-GFP:*Atp2b2*-mut1:lipo2000 into the inner ear of P1 *Atp2b2*^*Obl/+*^ mice. Four days after injection, the organ of Corti was harvested with the GFP positive sensory epithelium dissected from the rest of the cochlea so the transfected sensory epithelium was enriched. Following DNA extraction and PCR amplification of the genomic region harboring the *Obl* and WT alleles, we performed NGS analysis to assess the editing efficiency. We observed an indel rate of 0.4–1.2% in the *Obl Atp2b2* locus in the injected *Atp2b2*^*Obl/+*^ organ of Corti samples (Fig. 2a–c). No editing events were observed in the samples from uninjected contralateral control or WT *Atp2b2*^*+/+*^ mice injected with Cas9-GFP:*Atp2b2*-mut1 (Fig. 2b, c).

The relatively low in vivo indel rate in the samples enriched for hair cells could result from an overall low editing efficiency in hair cells and/or other chromosomal organizational changes including editing-induced large deletions which could not be detected by amplicon-based NGS analysis. If editing results in for instance large deletions harboring the mutant allele, our detection method to identify small indels could miss it. To assess if large deletions could be the result of editing undetected by NGS, we performed nested PCR on the DNA isolated from the RNP injected *Atp2b2*^*Obl/+*^ inner ear in vivo, using two pairs of nested flanking primers that were more than 1 kb apart in the genomic DNA centered around the cutting site defined by the sgRNA, to amplify both large (without large deletions) and small (with large deletions) DNA fragments (Fig. 2d). Nested PCR yielded large and small DNA fragments in the editing complex injected *Atp2b2*^*Obl/+*^ inner ear samples, while in injected WT and uninjected *Atp2b2*^*Obl/+*^ animals, nested PCR only yielded the full length *Atp2b2* locus without any small DNA fragments (Fig. 2e). NGS study of these small fragments (asterisks, Fig. 2e) revealed multiple deletions of DNA fragments of ~1.7 kb that harbored the *Obl* mutation in exon 24, with the ends of the deletions in the intronic regions flanking exon 24 (Fig. 2f). To

determine whether the large deletions were caused by off-target editing, we further amplified the fragment of the intronic sites (−997–1010 and +661–821 from the cut sites) (Supplementary Fig. 2f) and analyzed for small indels. We did not detect any indels at these intronic sites in injected WT samples and *Apt2b2*^*Obl/+*^ samples, which supports that the cause of the large deletion is unlikely a result of additional off-target editing. The possible reason for the large deletion may be the single on-target cut caused genetic lesion, which has been reported before[30]. All these results demonstrate that Cas9-GFP:*Atp2b2*-mut1 in vivo injection caused large deletions encompassing the *Obl* mutation.

In the *Atp2b2*^*Obl/+*^ inner ears, both WT and the *Obl* alleles were transcribed and translated specifically in the OHCs, with the protein products recognized by the antibody. We were thus unable to use immunolabeling to differentiate the *Atp2b2* from *Atp2b2*^*Obl*^ products as the readout for the editing event. As an alternative to study editing effect, we compared the ratio between WT and *Obl* alleles based on genomic DNA using purified hair cells. We performed the study by harvesting and dissecting the cochleae from RNP injected and uninjected *Atp2b2*^*Obl/+*^ inner ears, with the cochleae incubated with the vital dye FM1-43 to label hair cells. We then dissociated the cochleae so hair cells could be visualized and individually picked under an inverted fluorescent microscope for DNA extraction. We performed whole-genome amplification on DNA from isolated hair cells followed by PCR and NGS to identify WT and *Obl* alleles. In uninjected *Atp2b2*^*Obl/+*^ hair cells, we detected an *Obl*:WT ratio percentage of $52.57 \pm 3.47{:}47.5 \pm 3.48$ (mean ± sd). In injected inner ears, an *Obl*:WT ratio percentage of $46.18 \pm 2.55{:} 53.25 \pm 2.27$ (mean ± sd) was detected. There was 6.4% shift from the *Obl* towards the WT allele in hair cells after injection. We calculated that in injected ears, the editing efficiency on the *Obl* is ~21% (Fig. 2g). We further studied *Atp2b2* transcripts 1 week after injection to determine if editing resulted in a reduction of the *Obl* transcripts. We detected indels in the cDNAs of the *Obl* transcripts in injected animals but not in uninjected animals (Fig. 2h). Compared to the uninjected *Atp2b2*^*Obl/+*^ animals, we detected a significant decrease in un-modified Obl mRNA relative to un-modified WT mRNA in injected animals (Fig. 2h, i). As the sgRNA-*Atp2b2*-mut1 only targets the *Obl* allele, the ratio of un-modified Obl mRNA relative to WT mRNA has decreased by 20% (Fig. 2j). The data from DNA allele frequency and mRNA transcript study support that in as many as 20% of cochlear HCs, the *Obl* mutant allele was disrupted by editing, so the WT *Atp2b2* allele in the edited hair cells could restore HC function.

We next analyzed *Atp2b2* transcripts by NGS to determine if large deletions in exon 24 resulted in exon 24 skipped transcripts. No *Atp2b2* transcripts skipped exon 24 were detected, indicating that the transcripts without exon 24 were likely to be degraded rapidly. Combining the indels, the large deletions, the change in the allele frequency in favor of WT *Atp2b2* gene and a reduction in *Obl* mRNA, these results support that RNP delivery by Cas9:*Atp2b2*-mut1 sgRNA with Lipofectamine 2000 into the inner ear of *Atp2b2*^*Obl/+*^ mice in vivo results in conventional indels and uncommon large deletions around the cutting site, and as a result disrupts the *Obl* mutation in an estimated 21% of hair cells.

## In vivo editing on the *Obl* mutation of *Atp2b2* gene preserves outer hair cell morphology and survival

The *Atp2b2* gene is specifically expressed in the cochlear outer hair cells (OHC)[31]. Previous studies have shown that the *Atp2b2*^*Obl/+*^ mice exhibited progressive base to apex degeneration of hair cells, with OHC more affected than inner hair cells (IHC)[21]. To evaluate the effect of the Cas9:*Atp2b2*-mut1 sgRNA complex to target the *Obl* allele in the OHCs in vivo, we complexed the Cas9:*Atp2b2*-mut1 sgRNA with Lipofectamine 2000 and injected the mixture into the scala media of neonatal (P0-2) *Atp2b2*^*Obl/+*^ mice by cochleostomy, with the mouse

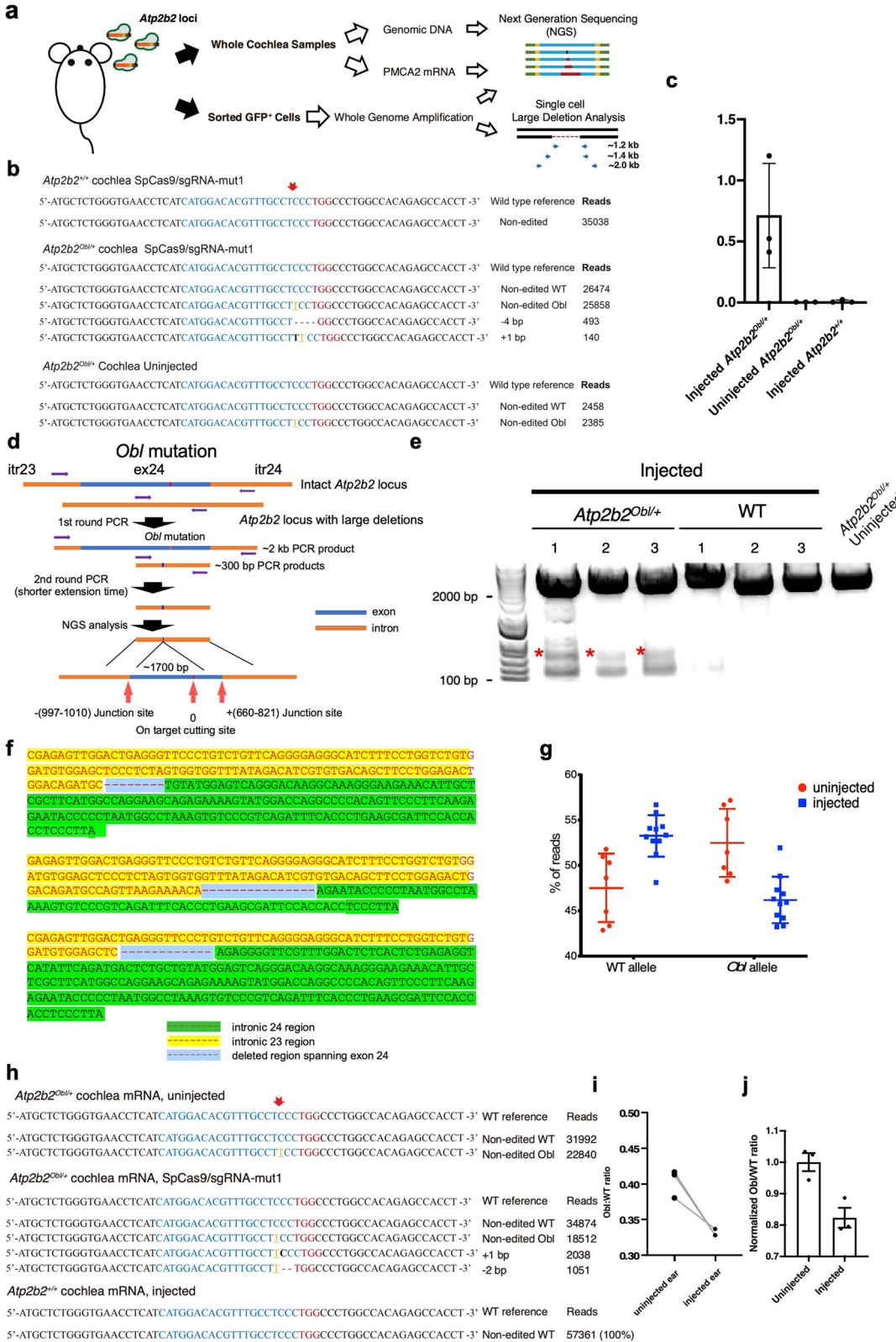

inner ears harvested at 2 months of age. We used an anti-Parvalbumin antibody to label all hair cells and an anti-PMCA2 antibody to visualize the distribution of PMCA2, the protein encoded by the *Atp2b2* gene, in the OHC. In uninjected *Atp2b2*[Obl/+] inner ear, there was a severe loss of HCs. In the high-frequency region (45.25 kHz), virtually all OHCs (PMCA2[+]) were absent with some missing IHC, whereas in other frequency regions, partial loss of OHCs was evident (Fig. 3a, b). In

contrast, in the injected *Atp2b2*[Obl/+] ears, most of the OHCs survived in the cochlea across all frequency regions at this age (Fig. 3a, b). There is a generally greater IHC loss in uninjected inner ear although it is not statistically significant (Fig. 3c). To further study hair cell structure, we performed scanning electron microscopy (SEM) to examine hair cell stereocilia of OHC and IHC in detail. Uninjected and Cas9:Atp2b2-mut1 sgRNA: Lipo2000 injected *Atp2b2*[Obl/+] mouse cochlea were

**Fig. 2 | In vivo gene editing by RNP injection in *Atp2b2 Obl* mutant mice.**
**a** Schematic overview of in vivo indel analysis experimental design. **b, c** Indel frequencies on the *Obl* allele from injected *Atp2b2^{obl/+}* or *Atp2b2^{+/+}* organ of Corti samples 4 days after injection of Cas9-GFP:*Atp2b2*-mut1:Lipo2000 or from uninjected *Atp2b2^{obl/+}* organ of Corti. The most abundant editing events by small deletions at the *Atp2b2* sequences, grouped by similarity, from the organ of Corti samples are shown on the right. The PAM sites were marked red. The *Obl* mutation is highlighted in yellow. The red arrow indicates the cutting site. Values and error bars are mean ± SD. **d** A schematic representation of nested PCR to detect large deletions resulting from injection of Cas9-GFP:*Atp2b2*-mut1:Lipo2000. Itr: intron; ex: exon. **e** Gel image of the nested PCR products from injected *Atp2b2^{obl/+}*, WT, and uninjected *Atp2b2^{obl/+}* samples. Red asterisks indicate small fragments of varying sizes used for NGS, and the upper bands are the -2 kb expected fragments. From injected WT and uninjected *Atp2b2^{obl/+}* samples, the nested PCR did not produce any smaller fragments. This was repeated independently for 3 times with similar results. **f** Representative reads from multiple NGS analyses of smaller nested PCR products showed deletions that spanned the entire exon 24 with the *Obl* mutation. **g** Comparison of WT and *Obl* allele frequency from isolated *Atp2b2^{ob}* hair cells showed a shift that increased the WT allele frequency and decreased the *Obl* allele frequency after in vivo injection of the Lipo:Cas9:*Atp2b2*-mut1 RNP complex compared to uninjected *Atp2b2^{obl/+}* hair cells. The dots represented independently purified hair cell groups from five animals. Values and error bars are mean ± SEM. **h** Indel profiles and read abundance at the mRNA level after injection of Lipo:Cas9:*Atp2b2*-mut1 RNP complex in *Atp2b2^{obl/+}* and WT mice. Indels were only detected in the mRNA of injected *Atp2b2^{obl/+}* but not in the mRNA of injected WT ears. **i** The ratio of un-modified *Obl* mRNA relative to un-modified WT mRNA in uninjected and injected *Atp2b2^{obl/+}* animals based on the NGS reads. Values and error bars are mean ± SD. **j** The normalized ratio of un-modified *Obl* and WT mRNA from (**i**). Values and error bars are mean ± SD, *n* = 3. Source data are provided as a Source data file.

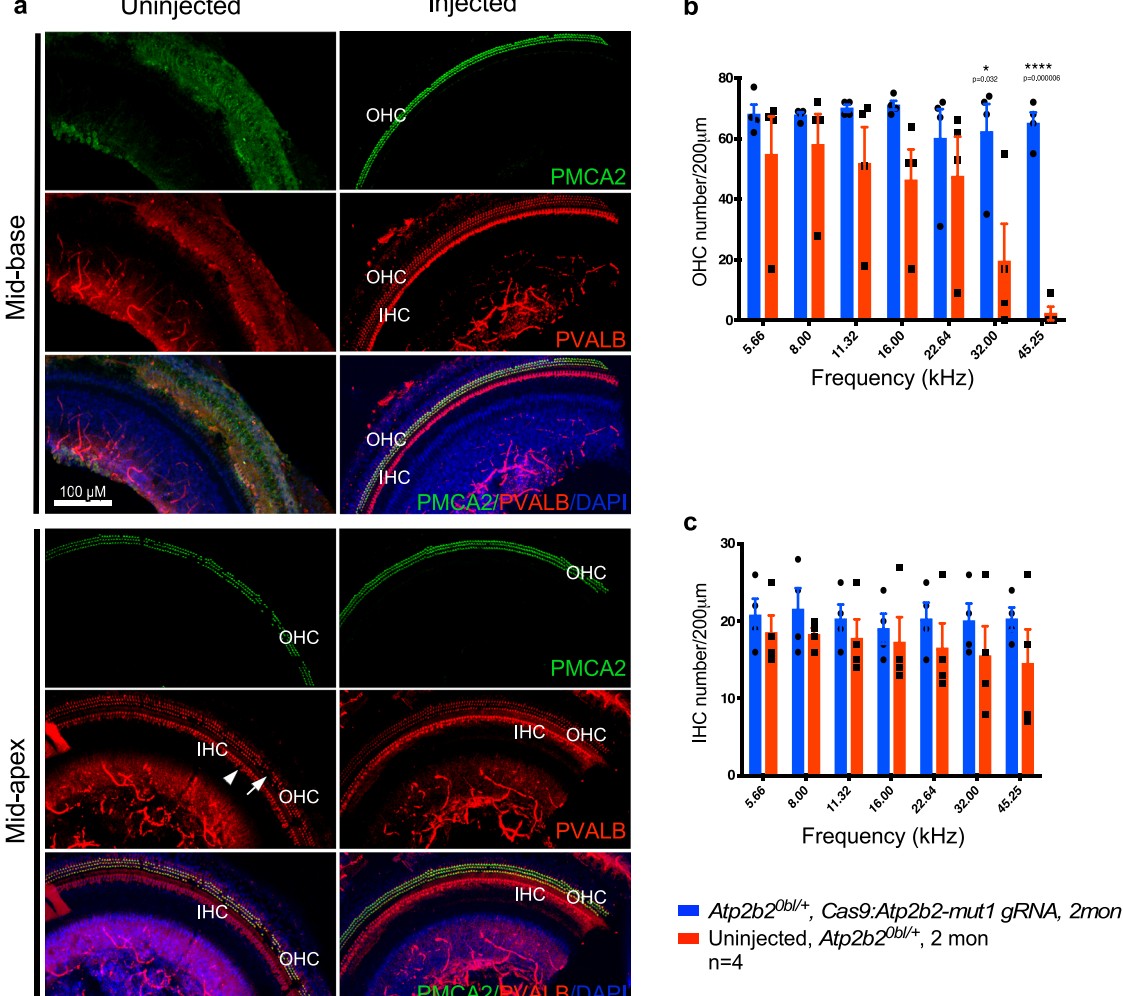

**Fig. 3 | In vivo lipid-mediated RNP delivery of Cas9:sgRNA complexes improves hair cell survival. a** Representative confocal microscopy whole mount images from uninjected (left) and Cas9:*Atp2b2*-mut1 sgRNA:Lipo2000 injected (right) *Atp2b2^{Obl/+}* cochleae at P1 with the cochleae harvested at 2 months of age. PMCA2 labeled OHC (PMCA2/PVALB) was detected in the injected *Atp2b2^{Obl/+}* cochlea and was missing in the uninjected contralateral control cochlea in the mid-base turn. In the mid-apical turn, OHCs were seen in injected and uninjected *Atp2b2^{Obl/+}* cochleae. **b, c** Quantification of OHC (**b**) and IHC (**c**) survival in *Atp2b2^{Obl/+}* mice 8 weeks after Cas9:*Atp2b2*-mut1 sgRNA:Lipo2000 injection (blue) compared to uninjected (red) contralateral ears across frequency regions. Values and error bars reflect the mean ± SEM of 4 mice (*n* = 4). Source data are provided as a Source data file. Paired t test was used for the analysis: \**p* < 0.05 and \*\*\*\**p* < 0.0001.

harvested at 4 weeks after injection, dissected, and then processed and imaged by SEM (Fig. 4a). The hair cells of uninjected *Atp2b2^{Obl/+}* mice cochlea showed the sign of degeneration including the missing shorter stereocilia in the outer hair cells and the disorganization of the stereocilia of the inner hair cells (left panel of Fig. 4b–e). The hair cells of injected *Atp2b2^{Obl/+}* mouse cochlea had organized and well-preserved stereocilia (right panel of Fig. 4b–e). These data demonstrate that Cas9/Atp2b2-mut1/Lipo2000 injection in vivo rescues *Atp2b2^{Obl/+}* hair cells by promoting HC survival and maintaining the stereocilia structure in *Atp2b2^{Obl/+}* mice.

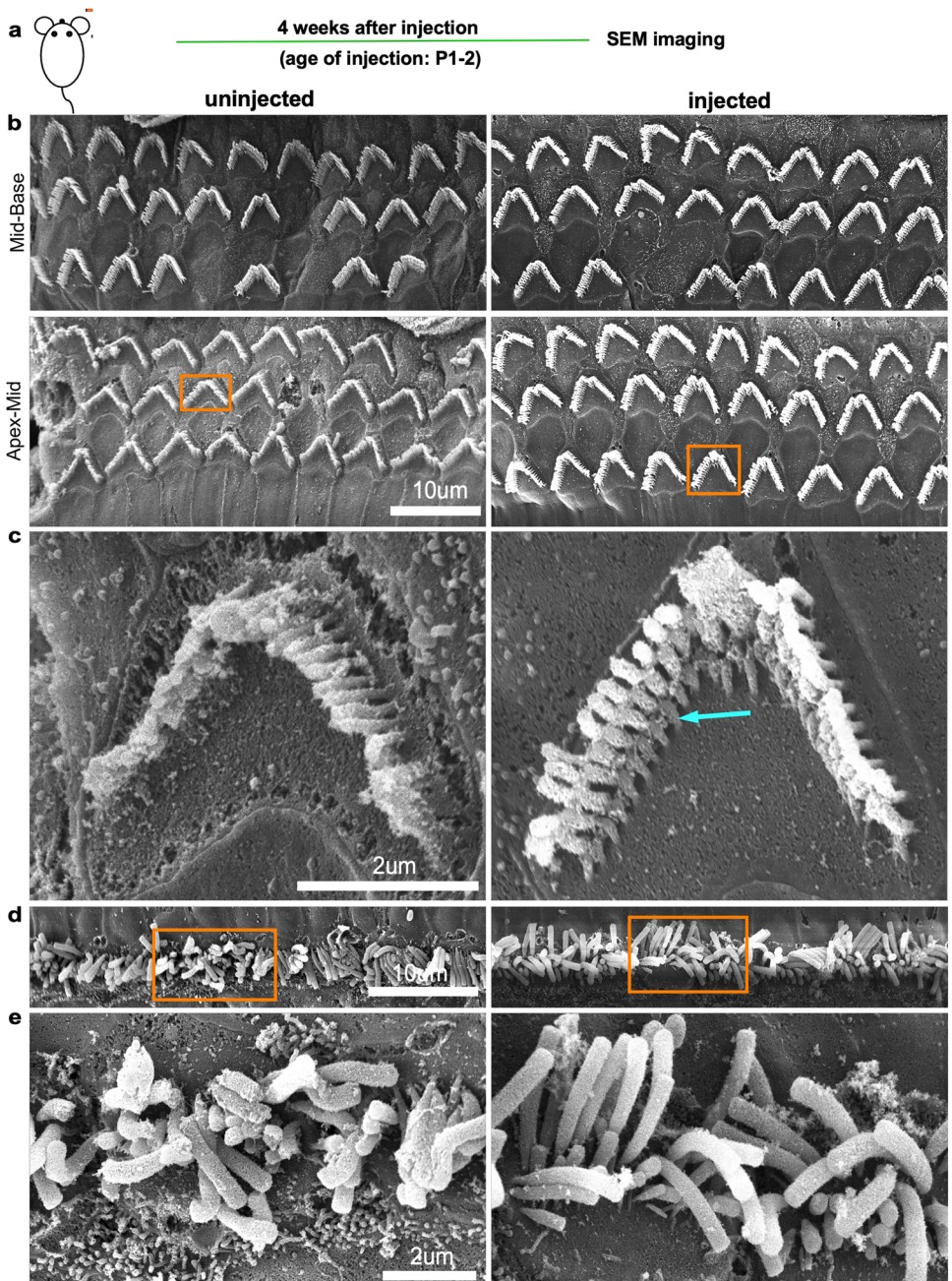

**Fig. 4 | In vivo lipid-mediated delivery of Cas9:sgRNA complexes restores hair bundle morphology. a** Experimental workflow. Samples were harvested, processed and imaged at 4 weeks after injection. **b** Images of scanning electron microscopy (SEM) of uninjected *Atp2b2^Obl/+* (left) and Cas9:*Atp2b2*-mut1 sgRNA: Lipo2000 injected *Atp2b2^Obl/+* (right) outer hair cell bundles. This was repeated independently for 3 times with similar results. Scale bar, 10 μm. **c** High-magnification images of outer hair cell bundles from the insets of (**b**) of uninjected *Atp2b2^Obl/+* (left) and Cas9:*Atp2b2*-mut1 sgRNA: Lipo2000 injected *Atp2b2^Obl/+* cochlea (right). The arrow points to shorter stereocilia in an outer hair cell from an injected ear that are missing from an outer hair cell from an uninjected ear. This was repeated independently for 3 times with similar results. Scale bar, 2 μm. **d** Images of scanning electron microscopy (SEM) uninjected *Atp2b2^Obl/+* (left) and Cas9:*Atp2b2*-mut1 sgRNA: Lipo2000 injected *Atp2b2^Obl/+* (right) inner hair cell bundles. This was repeated independently for 3 times with similar results. Scale bar, 10 μm. **e** High-magnification images of inner hair cell bundles from the insets of (**d**) of uninjected *Atp2b2^Obl/+* (left) and Cas9:*Atp2b2*-mut1 sgRNA: Lipo2000 injected *Atp2b2^Obl/+* cochlea (right). This was repeated independently for 3 times with similar results. Scale bar, 2 μm.

## In vivo editing on the *Obl* mutation of *Atp2b2* gene partially restores OHC function and recovers hearing

The goal of our study is to use editing to target the dominant OHC *Atp2b2 Obl* mutation to recover hearing. PMCA2 plays an essential role in the OHC function by regulating the $Ca^{2+}$ level to maintain homeostasis of the cochlea[19]. In the *Atp2b2^Obl/+* mice the *Obl* mutation has dominant effect that disrupts the function of PMCA2 and consequently the OHC function shown by the elevated thresholds of the distortion

product otoacoustic emissions (DPOAEs)[21]. The lack of OHC function results in profound hearing loss in the *Atp2b2^Obl/+* mice[21].

To evaluate the ability of the Cas9:*Atp2b2*-mut1 sgRNA complex to rescue OHC function, we injected lipid-mediated delivery of Cas9:*Atp2b2*-mut1 sgRNA complexes into neonatal (P0-P2) *Atp2b2^Obl/+* mice and analyzed the DPOAEs from 4 weeks onwards. Wild type 1-month-old C3H mice, with the same genetic background as the *Atp2b2^Obl/+*, were included for the measurement as additional controls.

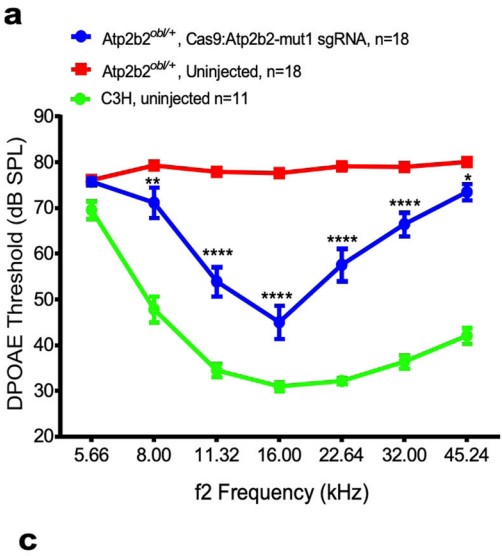

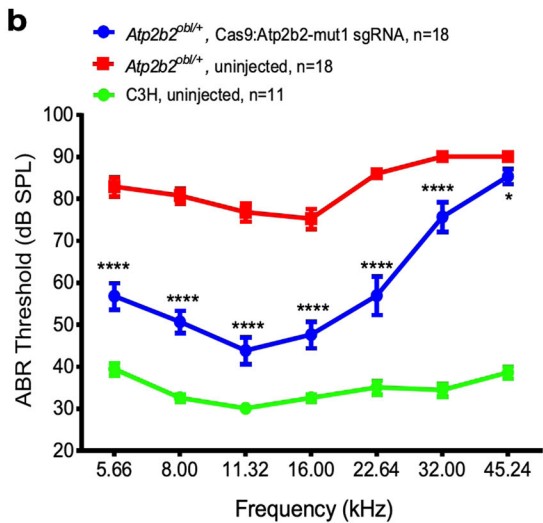

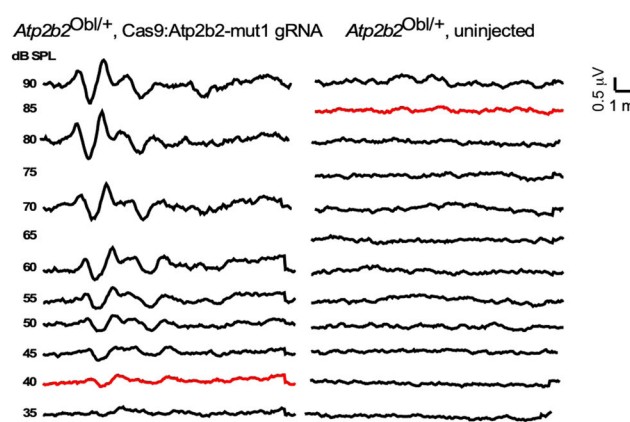

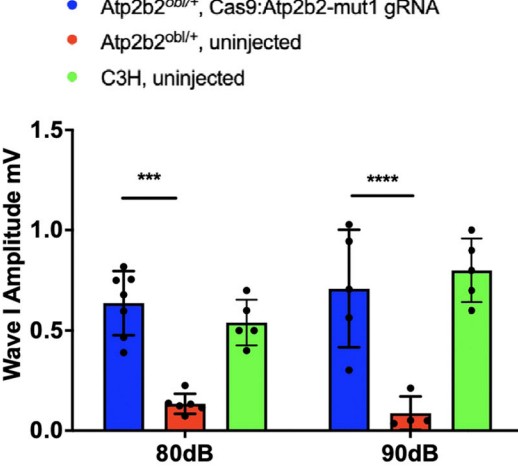

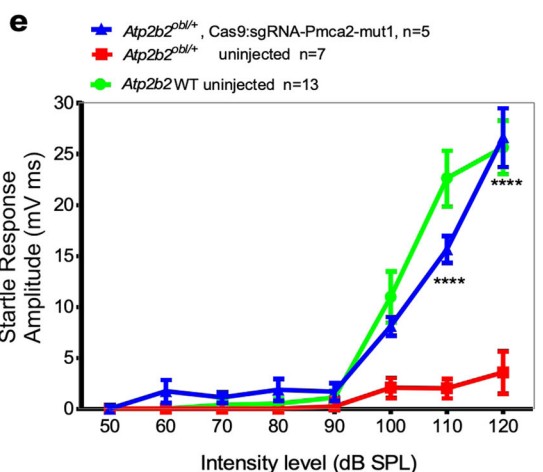

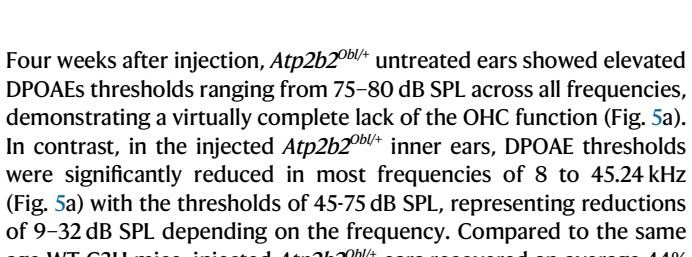

Four weeks after injection, *Atp2b2*^Obl/+ untreated ears showed elevated DPOAEs thresholds ranging from 75–80 dB SPL across all frequencies, demonstrating a virtually complete lack of the OHC function (Fig. 5a). In contrast, in the injected *Atp2b2*^Obl/+ inner ears, DPOAE thresholds were significantly reduced in most frequencies of 8 to 45.24 kHz (Fig. 5a) with the thresholds of 45-75 dB SPL, representing reductions of 9–32 dB SPL depending on the frequency. Compared to the same age WT C3H mice, injected *Atp2b2*^Obl/+ ears recovered an average 44%

of DPOAE thresholds vs uninjected ears across the frequencies of 8 to 45.24 kHz. At 16 kHz in the injected ears, DPOAE recovered to 68% of that in WT C3H ears. The frequency 5.66 kHz did not show a significant threshold change among injected, uninjected *Atp2b2*^Obl/+ and C3H inner ears.

The lack of OHC function leads to profound hearing loss in the *Atp2b2*^Obl/+ mice. To study the effect of Cas9:*Atp2b2*-mut1 sgRNA injection on hearing in *Atp2b2*^Obl/+ mice, we measured auditory

**Fig. 5 | In vivo lipid-mediated delivery of Cas9:sgRNA complexes improves OHC function, hearing and auditory behavioral function. a** DPOAE thresholds in *Atp2b2^Obl/+^* ears injected with Cas9:*Atp2b2*-mut1 sgRNA:Lipo2000 (blue), uninjected *Atp2b2^Obl/+^* ears (red), and wild-type C3H ears (green) after 4 weeks of age. **b** ABR thresholds in *Atp2b2^Obl/+^* ears injected with Cas9:*Atp2b2*-mut1 sgRNA:Lipo2000 (blue), uninjected *Atp2b2^Obl/+^* ears (red) (the thresholds for each individual mouse could be found in Source data file), and wild-type C3H ears (green) after 4 weeks of age. **c** Representative ABR waveforms illustrating a threshold of 40 dB (red trace, left) at 16 kHz in a Cas9:*Atp2b2*-mut1 gRNA injected *Atp2b2^Obl/+^* inner ear compared to the threshold of 85 dB (red trace, right) of the uninjected contralateral ear of the same mouse 4 weeks after injection. **d** Amplitudes of ABR Wave 1 at 16 kHz in Cas9:*Atp2b2*-mut1 sgRNA:Lipo2000-injected ears (blue), uninjected ears (red), and wild-type C3H ears (green) after 4 weeks of age. **e** Startle responses in Cas9:*Atp2b2*-mut1 sgRNA:Lipo2000 injected mice (blue), uninjected mice (red) and wild-type C3H ears (green) at 8 weeks post injection. Statistical tests were two-way ANOVA with Bonferroni correction for multiple comparisons: **$p < 0.01$, ***$p < 0.001$, and ****$p < 0.0001$. Values and error bars are mean ± SEM.

brainstem responses (ABRs), which assays the sound evoked neural output of the cochlea. Uninjected *Atp2b2^Obl/+^* inner ears showed elevations in cochlear neural responses at 4 weeks of age, with ABR thresholds ranging from 70–90 dB, compared with the 30–40 dB for wild-type C3H mice from 5.66 to 45.24 frequencies (Fig. 5b). In contrast, injected *Atp2b2^Obl/+^* ears showed significantly improved neural cochlear responses 4 weeks after injection, with lower ABR thresholds relative to uninjected ears (Fig. 5b, c). Significant hearing preservation was detected in all frequencies, with average ABR thresholds 26 dB lower in the treated ears than in uninjected contralateral ears. Compared with C3H control ears, injected *Atp2b2^Obl/+^* ears recovered hearing to 55% of the normal level. Greater wave I amplitudes were seen in the injected than in uninjected ears (Fig. 5d). As a behavioral measure of hearing preservation, we assessed acoustic startle responses 8 weeks after injection. Uninjected *Atp2b2^Obl/+^* mice showed profound attenuation of startle responses following stimulus at 100, 110, and 120 dB SPL individually, with the amplitudes 0–4 mV rms, compared to the 10–25 mV rms in the wild-type C3H mice (Fig. 5e). In contrast, injected *Atp2b2^Obl/+^* mice showed significant ($P < 0.001$) startle responses following stimulus at 110 and 120 dB SPL compared to the uninjected mice, with the startle response amplitudes that were similar or equal to those observed in the wild-type C3H mice (Fig. 5e). Thus, RNP delivery of the editing complex restored startle responses in *Atp2b2^Obl/+^* mice as the result of hearing recovery.

### Long-term partial recovery of OHC function and hearing
We continued the test of DPOAE and ABR of injected ears at 8 and 16 weeks after injection. We observed that the improvement in OHC activity achieved at 4 weeks was maintained in the treated ears at frequencies ranging from 11.32 to 22.64 kHz 8 weeks after the injection, with average DPOAE thresholds 22 dB SPL lower for the treated ears than untreated contralateral ears (Supplementary Fig. 3a). There was no statistical difference in DPOAE thresholds in injected ears between 4 and 8 weeks (Supplementary Fig. 3a). Smaller but significant ($P < 0.01$) improvement in DPOAE thresholds was still detected at 11.32 and 16 kHz 16 weeks after injection (Supplementary Fig. 3b). ABR threshold reductions were significant at the frequencies below 32 kHz at 8 weeks compared to that of uninjected ears, with an average reduction of 20 dB. The ABR threshold reduction was indistinguishable between 4 and 8 weeks post injection (Supplementary Fig. 3c). At 16 weeks post injection, the reduced ABR thresholds were detected at the frequency below 16 kHz with an average reduction of 14 dB in the injected compared to uninjected ears (Supplementary Fig. 3d). However, compared to 4 and 8 weeks post injection, the magnitude of the ABR threshold reduction at 16 weeks post injection was greatly reduced, an indication of diminished hearing recovery long term.

### Hearing recovery is gRNA specific and depends on the disruption of the *Obl* allele and the Cas9 cleavage activity
From the in vitro assay, the gRNA *Atp2b2*-mut1 showed the highest editing efficiency targeting the *Obl* allele, but not the WT allele. The gRNA *Atp2b2*-mut1 was chosen for the in vivo study. Two other gRNAs, *Atp2b2*-mut2 and *Atp2b2*-mut3, were less specific as they targeted the *Obl* and the WT alleles (Fig. 1b; Supplementary Fig. 1). To assess the ability of other less efficient *Atp2b2 Obl*-targeting sgRNAs in rescue of OHC function and hearing, we injected Cas9:*Atp2b2*-mut2:Lipo2000 complexes into P1 neonatal *Atp2b2^Obl/+^* mice. Four weeks after injection, we did not detect any improvement in the DPOAE or ABR thresholds of the *Atp2b2^Obl/+^* injected ears compared to the contralateral uninjected ears (Supplementary Fig. 4a, b). Thus the in vitro selection of the gRNA with the highest specificity on the mutant *Obl* allele predicated the in vivo functional study outcome.

To test whether amelioration of hearing loss requires the disruption of the mutant *Atp2b2 Obl* allele, we injected Cas9/sgRNA/lipid complexes targeting an unrelated gene (GFP) in *Atp2b2^Obl/+^* mice and observed no significant improvements in ABR or DPOAE thresholds (Supplementary Fig. 4c, d). In addition, to test whether preservation of cochlear function requires Cas9 nuclease activity, rather than transcriptional interference from Cas9 binding to *Atp2b2*, we injected *Atp2b2^Obl/+^* mice with catalytically inactive dCas9 complexed with *Atp2b2*-mut1 sgRNA and observed lower but not statistically different ABR or DPOAE thresholds in injected and uninjected ears (Supplementary Fig. 4e, f). Taken together, these results support that the hearing recovery in *Atp2b2^Obl/+^* mice is *Obl* allele specific and depends on Cas9 DNA cleavage activity.

### Double editing partially recovers the auditory function of digenic mutation origin in vivo
Increasingly, genetic hearing loss has been shown to be caused by digenic mutations, i.e., mutations in two different genes[32,33]. An editing approach to recover hearing would require editing that targets both mutations simultaneously. To establish the feasibility that RNP-mediated delivery of editing agents can target two separate mutations from different genes simultaneously, we bred the dominant mouse model *Atp2b2^obl/+^* with another dominant model *Tmc1^Bth/+^* to create a digenic dominant hearing loss model *Atp2b2^Obl/+^*; *Tmc1^Bth/+^* (double mutant), in which the *Obl* mutation primarily affects the OHC whereas the *Bth* affects the IHC. We performed liposome-mediated delivery of Cas9 complexed with individual *Atp2b2*-mut1 sgRNA, *Tmc1*-mut3 sgRNA or the mixture of both into the scala media of neonatal (P1) double mutant mice and tested the hearing 1 month after the injection. We have previously demonstrated that Cas9:*Tmc1*-mut3 sgRNA:Lipo2000 injection recovered hearing in the *Tmc1^Bth/+^* model[14]. Uninjected double mutant ears exhibited severely damaged auditory function, shown by large shifts in ABR and DPOAE thresholds across all frequencies (Fig. 6a–c). In the double mutant inner ears injected with the mixture of Cas9:*Atp2b2*-mut1/*Tmc1*-mut3 gRNAs, we detected significant hearing recovery for the frequencies below 32 kHz with an average reduction in the ABR thresholds of 10 dB ($n = 11$) (Fig. 6b). In the frequency range of 5.66 to 16 kHz, an average reduction of 14 dB was detected in ABR thresholds (Fig. 6b). Lower DPOAE threshold at 16 kHz ($p < 0.05$) was also detected in the injected inner ears (Fig. 6c). Injections of Cas9 complexed with single gRNA targeting the *Obl* or the *Bth* mutation individually failed to recover hearing (Fig. 6e–g). Taking together, these results suggest that a limited recovery from hearing loss of digenic mutations requires two specific sgRNAs targeting the two mutations at the same time.

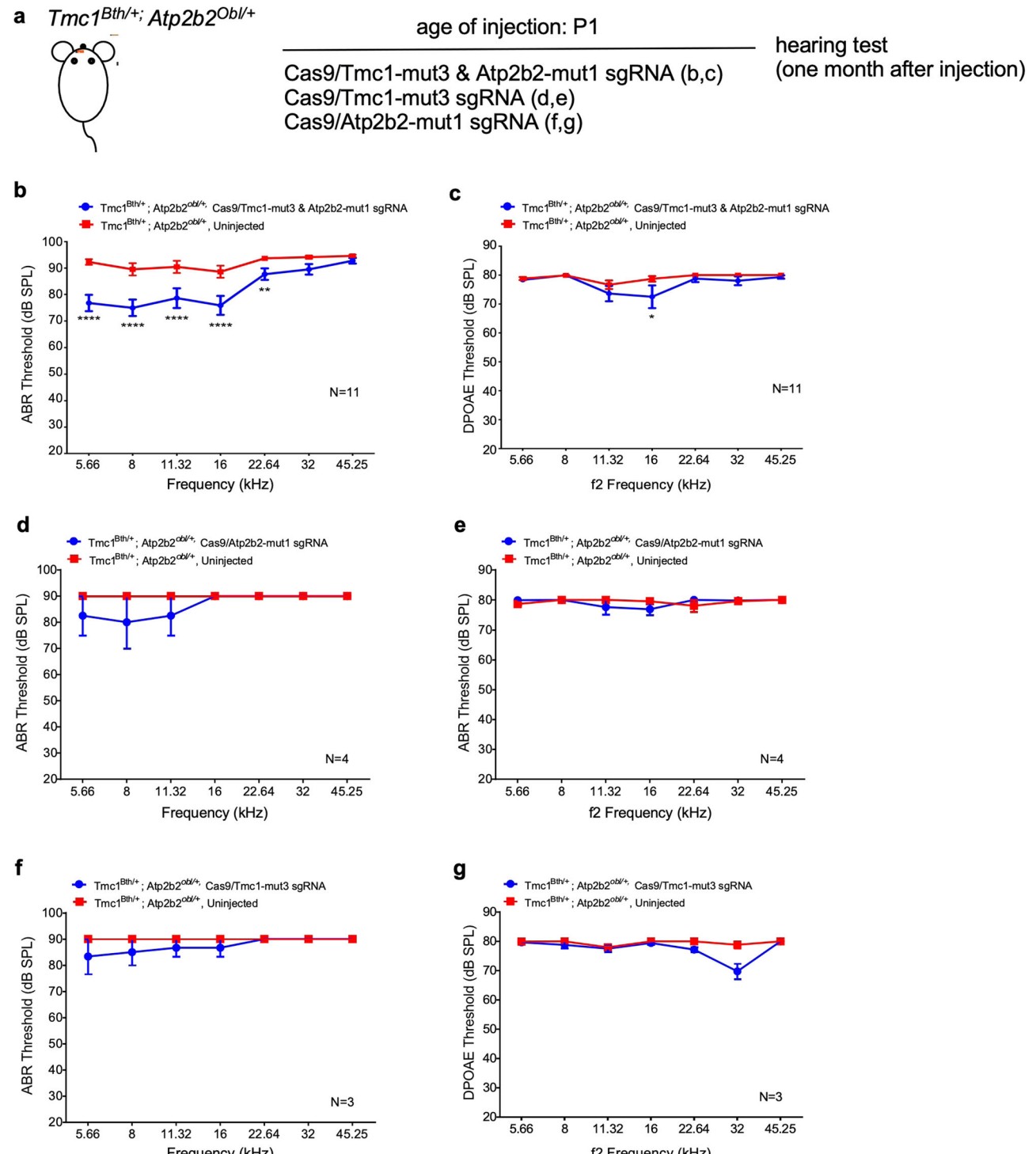

**Fig. 6 | In vivo gene editing partially recovers hearing of digenic mutation origin.** **a** A schematic diagram of the experimental design. **b** ABR and **c** DPOAE thresholds in $Atp2b2^{Obl/+}$: $Tmc1^{Bth/+}$ ears injected with Cas9:Atp2b2-mut1:Tmc1-mut3:Lipo2000 (blue) and uninjected $Atp2b2^{Obl/+}$: $Tmc1^{Bth/+}$ ears (red) at 4 weeks post injection. **d** ABR and **e** DPOAE thresholds in $Atp2b2^{Obl/+}$: $Tmc1^{Bth/+}$ ears injected with Cas9:Atp2b2-mut1 sgRNA:Lipo2000 (blue) and uninjected $Atp2b2^{Obl/+}$:$Tmc1^{Bth/+}$ ears (red) at 4 weeks of age. **f** ABR and **g** DPOAE thresholds in $Atp2b2^{Obl/+}$:$Tmc1^{Bth/+}$ ears injected with Cas9:Tmc1-mut3:Lipo2000 (blue) and uninjected $Atp2b2^{Obl/+}$: $Tmc1^{Bth/+}$ ears (red) at 4 weeks post injection. $**p < 0.01$, $***p < 0.001$, and $****p < 0.0001$. Values and error bars reflect mean ± SEM.

## Discussion

PMCA2 is essential to OHC function and is required for hearing[18–21]. Outer hair cells are known to be the most vulnerable components of the auditory system and their malfunction is implicated in sensory hearing loss pathologies. Without proper functional OHC, hearing sensitivity and frequency selectivity are impaired[34]. Mutations in the

$Atp2b2$ have been associated with dominant progressive hearing loss in humans[18]. An $Atp2b2$ variant has also been shown to be a modifier that contributes to the severity of hearing loss[20]. In mice, $Atp2b2$ encodes the resident $Ca^+$ pump of hair cell stereocilia. The $Obl$ mutation causes dysfunction in the transmembrane domain 6, which significantly reduces the nonactivated $Ca^{2+}$ exporting ability. The dysfunction of

Ca$^{2+}$ export by the mutated PMCA2 pump causes progressive hearing loss[21]. Given the importance of PMCA2 in the OHC function and hearing loss, and the dominant transmission pattern, the *Obl* mouse model is a good model to study using editing to disrupt the mutation and to recover hearing.

We have previously demonstrated the feasibility of liposomal-mediated RNP delivery of editing agents in hearing recovery in a mouse model of human genetic hearing loss DFNA36[14]. In that model, however, the mutation Beethoven in the *Tmc1* gene primarily affects inner hair cells. Furthermore, DPOAE thresholds in the treated inner ears were elevated compared to uninjected control ears. This raised a question if our approach is suitable to target OHC to edit mutations for hearing recovery. In addition to PMCA2, OHC mutations have been associated with the most common forms of dominant genetic hearing loss, including KCNQ4[35].

The current work shows that liposome-mediated delivery of Cas9-sgRNA RNP complexes in the inner ear leads to editing that abolishes the *Obl* mutation in the *Atp2b2* gene, restoring the functions of OHCs with improved cell survival and recovery of hearing in dominant genetic mouse model Oblivion with OHC deficit. The recovery of auditory function quantified here using physiological and behavioral assays suggests that transient lipid-mediated delivery of RNP complexes may be a good therapeutic strategy to target OHC for editing, and recover hearing of dominant genetic OHC dysfunction. Combined with our previous work on editing for hearing recovery in the inner hair cells[14], it supports that liposomal-mediated RNP delivery can be applied to recover hearing in dominant hearing loss models of inner and/or outer hair cell origin.

Our study shows recovery in DPOAE and ABR thresholds after injection of editing agents in the *Atp2b2*$^{Obl/+}$ inner ear, with reductions of 9−32 dB SPL depending on the frequency. Based on our results for 8 and 16 weeks after injection, the treatment effect starts to diminish at 16 weeks post injection. While the underlying mechanism of the diminished hearing recovery is unknown, one hypothesis is that the duration of hearing recovery is related to the equilibrium between edited and unedited hair cells. In our study, we detected editing in ~20% of hair cells, leaving more hair cells unedited. If unedited hair cells start to degenerate, the edited hair cells may not be able to sustain the cochlear function due to the death of unedited HC which may impact both the cochlear structure and function. In addition, the diminished hearing recovery could be gene specific in that the production of 50% of normal PMCA2 protein after editing of the mutant allele may not be sufficient to maintain the health and function of the cochlea long-term. To sustain hearing recovery, improved editing efficiency in more OHC will likely be critical. Another possibility is that the mutant hair cells may have started to degenerate at the time of injection (p0-p2) whereas editing delays the degeneration process. Indeed, significant elevation of DPOAE and ABR thresholds was seen at 1 month of age in the *Atp2b2*$^{obl/+}$ ear, suggesting hair cells may have undergone some degeneration at an early stage. In this case, intervention at an even earlier age (e.g., in the embryos) may improve the recovery effect. The significant difference between early and later inventions has been illustrated in most hearing recovery studies in which the treatment was only effective when the injection was done in the neonatal stage whereas it would fail if the intervention were done even a few days later (e.g., p6)[36].

The hearing recovery is likely due to efficient editing at the *Obl* allele. By NGS and indel analysis from the *Atp2b2*$^{Obl/+}$ adult primary fibroblasts treated by nucleofection of the RNP complex in vitro or from the *Atp2b2*$^{Obl/+}$ inner ears injected with the complex in vivo, the overall indel rate is relatively low (~1%), which suggested that the analysis underestimated the editing efficiency. A few factors may have contributed to the low indel rate detected. First, editing in the *Atp2b2* locus in adult fibroblasts may be intrinsically less efficient than in other rapidly dividing cell type. After we used the transposon to introduce the DNA fragment harboring the *Obl* mutation into an organ of Corti cell line that is rapidly dividing, we detected an indel rate of 40% after nucleofection of the same RNP complex, suggesting cell type influences editing efficiency. Second, in the cochlear samples collected from in vivo study for indel analysis, the hair cells only consist of a small portion of all the cochlear cells. Third, we identified large deletions from the samples of in vivo studies, which a standard indel analysis failed to detect. As we did not exhaustively characterize large deletions, our selective PCR-based assay is likely to miss other deletions or DNA organizational changes.

Previous studies by us and others have shown that indel analysis using whole cochlea will substantially underestimate the editing efficiency primarily due to the scarcity of hair cells[14,15]. By single-cell whole-genome amplification of purified hair cells for NGS, we detected a reduction in the allele frequency ratio from *Obl* to the WT after RNP delivery. While it is expected that in uninjected *Atp2b2*$^{Obl/+}$ inner ear, the allele frequency ratio of *Obl*:WT should be 50%:50%. The detected frequency ratio of *Obl*:WT, however, was 53%:47%, which was likely the result of the bias introduced during genome amplification that favored the *Obl* allele. After the injection, the *Obl* allele decreased by over 6.4% compared to untreated hair cells, which supports editing in 12.8% of the *Obl* allele. The lower *Obl* allele frequency after editing is consistent with the data that only the *Obl* allele was edited by RNP delivery of Cas9-*Atp2b2*-mut1 in vitro and in vivo (Fig. 2a; Supplementary Fig. 2c). In injected *Atp2b2*$^{Obl/+}$ ears, the *Obl* transcripts decreased by 20% compared to uninjected ears, further demonstrating the editing that targets the *Obl* mutation (Fig. 2i, j). Measuring the *Obl* vs WT allele frequency is also a better representation of editing as larger deletions could be reflected by a shift in allele frequency but will be missed by indel analysis based on amplicons. Future improvement in unbiased genome amplification should facilitate the study of allele frequency in small number of cells, which in turn may offer a better assessment of editing efficiency that may be underestimated due to unconventional editing events including large deletions.

Liposomal mediated RNP delivery showed high-specificity and low off-target effect. In in vitro and in vivo samples with RNP delivery of *Atp2b2*-mut1, we only detected editing at the *Obl* locus. We assessed editing at off-target loci that could be modified by plasmid or RNP-based editing using computational prediction[29] and the GUIDE-seq method[27]. Among the top five software suggested off-targets, only one was shown to be edited at a low efficiency of 0.3% after plasmid nucleofection, whereas RNP delivery did not produce any indel. GUIDE-seq analysis revealed the only cleaved site was the *Atp2b2 sequence* after RNP delivery.

The in vivo recovery of OHC and auditory functions correlated with in vitro gRNA selection, i.e., the gRNA-mut1 with the highest allele specificity for the *Obl* locus produced the best outcome, whereas gRNA-mut2, despite its editing at the *Obl* locus (Fig. 1c), did not rescue either hearing or OHC function (Supplementary Fig. 4a, b). The study supports that a thorough in vitro selection of sgRNA could significantly impact in vivo results. The high specificity is demonstrated by the comparison of editing on the *Obl* and WT alleles. We showed that both in vitro and in vivo there is no editing on the WT allele and all editing is at the *Obl* allele.

While a majority of on-target editing consists of indels of less than 20 bp, recent studies have shown that the double-strand breaks induced by CRISPR can also lead to large deletions, complex DNA rearrangements and chromosomal truncations in cell lines[30,37] and zygotes[38]. Our results are in line with these studies and show for the first time that large deletions due to editing also occur in vivo in the inner ear after the administration of CRIPSR/Cas9 editing agents. Future development of CRISPR-based therapy requires careful evaluation of such events and determines if they lead to interruption of important exons or genes.

We further provide evidence that the liposome RNP delivery system has the potential to be applied to partially improve hearing in mice carrying separate mutations in two genes. Genetic hearing loss due to digenic mutations is well documented[20,22–25]. With the identification of increasing number of deafness genes, it is likely future intervention by editing therapy needs to target more than one mutation simultaneously. We took advantage of two dominant genetic hearing loss models, $Atp2b2^{Obl/+}$ and $Tmc1^{Bth/+}$ to create a double mutant model ($Atp2b2^{Obl/+}/Tmc1^{Bth/+}$) with profound hearing loss at 1 month. It is significant that injection of Cas9 complexed with the double gRNAs of $Atp2b2$-mut1 and $Tmc1$-mut3 recovered hearing shown by ABR thresholds significantly, whereas no hearing was recovered by a single gRNA delivery. The results suggest that in some hair cells both mutant $Obl$ and $Bth$ alleles have been edited that led to hearing recovery. This result is interesting given the fact that the delivery of each gRNA alone resulted in the editing efficiency of ~10% for Bth[14] and for $Obl$, thus a double editing of two mutations in the same hair cell is calculated around 2%. One possible explanation for hearing recovery in the double mutant mice is that the editing by the two gRNAs is not a random event, i.e., a hair cell may be preferentially edited by both gRNAs that gives rise to a skewed outcome. It is also possible that editing efficiency by each gRNA is higher than our data that was estimated to be the baseline for editing efficiency, which may contribute to higher number of hair cells being edited at both alleles. Another explanation is that editing in one hair cell may restore some function whereas collectively, there is an additive effect when both gRNAs were delivered. Despite the hearing recovery in the two-gRNA injected digenic mutant inner ears is much smaller compared to each gRNA injected to $Atp2b2^{Obl/+}$ or $Tmc1^{Bth/+}$ individually, the partial hearing improvement observed after RNP delivery supports further development to improve the efficacy, which should open the possibility for the approach as a potential treatment of hearing loss of multigenic origin. Our work provides evidence on the recovery of hearing based on the auditory function tests of ABR and DPOAE. We did not address how hearing recovery may affect mouse behavior. For the digenic mouse model, it is likely that more efficacious hearing recovery (e.g., less than 70 dB in ABR thresholds) will be needed to benefit the mice in behaviorally significant ways.

Our work demonstrates the feasibility of RNP delivery of editing agents into the inner ear to target outer hair cell mutations and recover hearing in the $Atp2b2^{Obl/+}$ mice with the effect that is reduced long term. We have shown that RNP delivery is suitable for the intervention of dominant mutations in the auditory hair cells. Combined with the evidence of the partial recovery effect in the digenic mouse mutants with hearing loss, we have further widened the path for future development of editing base therapy for genetic hearing loss.

## Methods

### In vitro transcription of sgRNAs
Linear DNA fragments containing the T7 promoter binding site followed by the 20-bp sgRNA target sequence were obtained by PCR using Q5 High-Fidelity DNA Polymerase (NEB) with the primers listed in the supplementary information and 40 ng of pFYF1320 (EGFP sgRNA expression plasmid) as a template according to the manufacturer's instructions. PCR products were purified using QUIAquick PCR purification kit (Qiagen) and transcribed in vitro using the T7 High Yield RNA Synthesis Kit (NEB) according to the manufacturer's instructions. sgRNA products were purified using the MEGAclear Transcription Clean-Up Kit (Ambion), quantified by Nanodrop and stored at −80 °C.

### In vitro DNA cleavage assay
$Atp2b2^{+/+}$ and $Atp2b2^{Obl/Obl}$ genomic DNA was isolated from mouse tail snips using the Agencourt DNAdvance Genomic DNA Isolation Kit (Beckman Coulter) according to the manufacturer's instructions. 956-bp $Atp2b2$ DNA fragments were amplified from the genomic DNA and

used as substrates for the in vitro cleavage reaction (Forward: GGACACTGAACCCCTGAGA; Reverse: GCCGAGAAAGGAGCTGACAT). Typically, the $Atp2b2$ DNA fragment (150 nM) was incubated for 15 min at 37 °C with 300 nM of purified Cas9 protein and sgRNA in Cas9 cleavage buffer (20 mM HEPES pH 7.5, 150 mM KCl, 0.5 mM DTT, 0.1 mM EDTA with10 mM MgCl2) in a total volume of 20 μL in each reaction. Reactions were quenched by adding 500 μL of PB wash buffer (Qiagen), purified on a QIAprep spin column and eluted in 20 μL 1X TE buffer. 10 μL of each reaction were loaded onto a 2% agarose gel and electrophoresed to separate starting DNA and cleaved products. Cas9-induced cleavage bands and the uncleaved band were visualized on an ChemiDoc MP and quantified using ImageJ software. The peak intensities of the cleaved bands were analyzed as previously described[39].

### Construction of $Obl$-OC1 cell line
An $Atp2b2$ gene fragment (2-kb) harboring the $Obl$ mutation was amplified by PCR from $Atp2b2^{Obl/Obl}$ mouse genomic DNA using primers (Forward: CCCAAGCTTCGAGAGTTGGACTGAGGGTT; Reverse: ATAAGAATGCGGCCGCTAAGGGAGGTGGTGGAATCG). The PCR products were ligated into the restriction sites (HindIII and NotI) in the PiggyBac donor backbone (PB-CAG-mNeonGreen-P2A-BSD-polyA)[40]. The constructed donor plasmid was co-transfected with PiggyBac transposon vector (PB210PA, System Biosciences) in HEI-OC1 cells. Cells with inserted $Atp2b2$ gene fragments carrying the $Obl$ mutation were cultured and selected in the medium containing 10 μg/mL Blasticidin. Successful insertion was confirmed by PCR (Forward: GCCATGAACAAAGGTTGGCT; Reverse: GAGAGTCCAAACGAACCCCT) and sequencing analysis.

### Delivery of protein complex or plasmids by nucleofection
Transfection programs were optimized following manufacturer's instruction (DS120, SG Cell Line 4D-Nucleofector™ X Kit). 400 ng of pmaxGFP Control Vector (LONZA) was added to the nucleofection solution to assess nucleofection efficiencies in HEI-OC1, $Obl$-OC1 cells and primary fibroblasts. To deliver plasmids, cells were transfected using 1000 ng Cas9 plasmid (pCas9), 500 ng sgRNA plasmid (pAtp2b2-mut1 sgRNA). To deliver protein complex, purified sgRNA was incubated with Cas9 protein for 5 min before transfection. Media was replaced ~16 h after nucleofection and cells were harvested for genomic DNA extraction after ~96 h.

### General in vivo experiments
All in vivo experiments were carried out in accordance with NIH guidelines for the care and use of laboratory animals and were approved by the Massachusetts Eye & Ear Infirmary IACUC committee. Isogenic heterozygous $Obl/+$ mice maintained on a C3HeB/FeJ (C3H) background were obtained from Wellcome Trust Sanger Institute. $Atp2b2^{Obl/+}$ mice were bred with $Tmc1^{Bth/Bth}$ to obtain $Pmca2^{Obl/+}$ $Tmc1^{Bth/+}$ mice.

### Microinjection into the inner ear of neonatal mice
A total of 112 $Atp2b2^{Obl/+}$ mice (P0-2) of either sex were used for injections. The mice were randomly assigned to the different experimental groups, and at least three mice were injected in each group. Both surgical procedures and injections were performed as described previously[14]. Briefly, 25 μM of Cas9 and sgRNA was mixed with Lipofectamine 2000 for 20 min at room temperature, and injected into the scala media of the cochlea at three different sites (base, middle and apex-middle turn). The volume for each injection was 0.3 μl with a total volume of 0.9 μl per cochlea. The release rate was 69 nl/min, controlled by a MICRO4 microinjection controller (WPI).

### Acoustic testing
Auditory brainstem responses (ABR) and distortion product otoacoustic emissions (DPOAE) were performed in a soundproof chamber

as described previously[14]. The acoustic tests were performed in mice up to 4 months old anesthetized intraperitoneally with a mix of xylazine (10 mg/kg) and ketamine (100 mg/kg). Three subcutaneous needle electrodes placed at vertex, the ventral edge of the pinna and the tail (ground reference), were used for the recordings. Stimuli consisted in 5 ms tone pips (0.5 ms rise–fall with a cos2 onset, delivered at 35/s) presented at frequencies 16, 22.64, 32, and 45.25 kHz and delivered in 10 dB ascending steps from 20 to 90 dB (Sound Pressure Level, SPL). The response was amplified 10,000-fold, filtered (100 Hz–3 kHz passband), digitized, and averaged (1024 responses) at each SPL. Following visual inspection of stacked waveforms, ABR threshold was defined as the lowest SPL level at which any wave could be detected. Wave 1 amplitude was defined as the difference between the average of the 1-ms pre-stimulus baseline and the Wave 1 peak (P1). The cubic distortion product for DPOAE measurements was quantified in response to primaries f1 and f2. The primary tones were set as previously described[14]. Threshold was computed by interpolation as the f2 level required to produce a DPOAE at 5 dB SPL.

## Acoustic startle reflex

Mice were placed into a small, acoustically transparent cage resting atop a piezoelectric force plate in a sound-attenuated booth. Acoustic stimuli and amplified force plate signals were encoded by a digital signal processor (Tucker-Davis Technologies, RX6) using LabView scripts (National Instruments). Mice were placed in silence for 5 min to acclimate to the test environment before real measurements. A 16-kHz tone was presented at randomized intervals from an overhead speaker (50 dB to 120 dB SPL, 20 ms duration with 0 ms onset and offset ramps). Twelve repetitions were recorded for each of the intensities per test subject. Startle response amplitude was measured as the root mean square (RMS) voltage of the force plate signal shortly after sound presentation.

## Immunohistochemistry and histology

Injected and non-injected cochleae were removed after animals were sacrificed by $CO_2$ inhalation. Temporal bones were fixed in 4% paraformaldehyde at 4 °C overnight, and then decalcified in 120 mM EDTA for at least 1 week. When the bone was decalcified, the organ of Corti was dissected in pieces for whole-mount immunofluorescence. Nonspecific labeling was blocked with blocking solution (PBS with 8% donkey serum and 0.3–1% Triton X-100) for 1 h at room temperature and followed by overnight incubation at room temperature with the primary antibody 1:300 mouse anti-Parvalbumin (Sigma P3088), 1:200 rabbit anti-pmca2 (PA1-915 ThermoFisher scientific). Tissues were incubated with the secondary antibody for 1 h after three rinses with PBS. All Alexafluor secondary antibodies were from Invitrogen: donkey anti-rabbit Alex488 (A21206) and donkey anti-mouse Alex594 (A32744) was used as a 1:500 dilution. All specimens were mounted in ProLong Gold Antifade Mountant medium (P36930, Life Technologies). Confocal images were taken with a Leica TCS SP8 microscope using a 20X or 63X glycerin-immersion lens, with or without digital zoom. We acquired z-stacks by maximum intensity projections of z-stacks for each segment by imageJ (NIH image), and composite images showing the whole cochlea.

## Hair cell isolation for DNA sequencing

*Atp2b2^Olb/+* mice were injected with Cas9-GFP:Pmca2-mut1 sgRNA:Lipofectamine 2000 at P1 and were euthanized at P5. Cochleae were harvested with the sensory epithelia (GFP⁺) dissociated using needles under the microscope (Axiovert 200M, Carl Zeiss), and immersed in 1 μM FM 1-43FX (PA1-915, ThermoFisher) dissolved in HBSS (ThermoFisher) for 30 s at room temperature in the dark. The sensory epithelia were then transferred to 100 μl TrypleE Express Enzyme (12604013, ThermoFisher) with the GFP⁺ cells isolated using a 1 μl pipette, Isolated hair cells were subjected to whole-genome

amplification by MALBAC Single Cell WGA Kit (YK001A, Yikon Genomics) to isolate DNA for NGS analysis.

## High-throughput DNA sequencing of genomic DNA samples

Treated cells or tissues were collected after four days and genomic DNA was isolated using the Agencourt DNAdvance Genomic DNA Isolation Kit (Beckman Coulter) according to the manufacturer's instructions. On-target and off-target genomic regions of interest were amplified by PCR with flanking primers (Forward: CCTCTCAAG GCTGTGCAGATGCT; Reverse: CCACGAAGAGCAGGGTGAAGATGA). PCR amplification was carried out with Q5 High-Fidelity DNA Polymerase (NEB) according to the manufacturer's instructions using 250 ng genomic DNA as a template. PCR products were purified using QUIAquick PCR purification kit (Qiagen). Samples were sequenced and analyzed to detect CRISPR variants from NGS reads using a custom algorithm developed by the Massachusetts General Hospital Center for Computational and Integrative Biology DNA Core. Individual SEQ files containing the variants were visualized by SnapGene.

## GUIDE-seq analysis

To identify off-target sites, the GUIDE-seq method was performed essentially as previously described (GUIDE-seq ref) with the following modifications. Briefly, 100 nmol Cas9 (IDT, Cat # 1081059), 800 ng sgRNA (Synthego Co, Menlo Park, CA), and 1 pmol of the double-stranded oligodeoxynucleotide (dsODN) were nucleofected into *Obl*-OC1 cells. A sample nucleofected with dsODN only served as a negative control. Cells were harvested for genomic DNA extraction after 5 days and ~400 ng of genomic DNA for each sample was sheared acoustically using a Covaris E220 sonicator to an average of 500 bp in 130 μL TE buffer. The GUIDE-seq sequencing libraries were prepared and sequenced on an Illumina Miseq as previously described[27], with reads were mapped to the mouse reference genome (GRCm38). Off-target data analysis was initially performed with the standard pipeline, mapping the start position of the amplified sequences using a 10-bp sliding window, then retrieving the reference sequence around the site. Given the size of some of the deletions, the number of base pairs used as the flanking sequence was also increased to up to 10,000 bp. The retrieved sequences were aligned to the Cas9 target sequence using a Smith-Waterman local-alignment algorithm. The negative control sample treated with the dsODN but no Cas9 or sgRNA was used to assess background.

## Scanning electron microscopy

After cochlea dissection, tissues were harvested and immersed in 2.5% glutaraldehyde in 0.1 M cacodylate buffer (EMS) supplemented with 2 mM $CaCl_2$ for 1.5–2 h at room temperature on a tissue rotator. Samples then were washed three times with distilled water. After that, samples were rinsed three times for 10 min with 0.1 M sodium cacodylate buffer and treated with 1% osmium tetroxide (one) for one hour, then a thorough rinse three times for 10 min each with distilled water. And samples were treated with saturated thiocarbohydrazide (two) in distilled water for 30 min. Repeat treatment was followed by an additional 1% osmium tetroxide step, making a treatment order of one-two-one-two-one. Then, samples were transferred to 20 ml scintillation vials for dehydration (in 2 ml distilled water) with ethanol, adding 50 μL of 100% ethanol to the vial, followed by doubling the added volume every 10 min. When the vial was full, samples were transferred to 100% ethanol, dried from liquid $CO_2$ (Tousimis Autosamdri 815) in terms of critical point. Then, samples were mounted with a carbon tape on aluminum specimen stubs, spatter-coated with 4.5 nm of platinum using Leica EM ACE600 and operated with Hitachi S-4700 SEM.

## Statistical analysis

Statistical analyses were performed by two-way ANOVA with Bonferroni corrections for multiple comparisons for ABRs and DPOAEs, and

by Student t-test for quantification of hair cell survival using the Prism 6.0 statistical analysis program (GraphPad).

## Reporting summary

Further information on research design is available in the Nature Portfolio Reporting Summary linked to this article.

## Data availability

High-throughput sequencing data have been deposited in the NCBI Sequence Read Archive database under accession code SUB13659056 and Bioproject PRJNA994108. All data supporting the findings described in this manuscript are available in the article and in the Supplementary Information and from the corresponding author upon request. Source data are provided with this paper.

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

## Acknowledgements

This work was supported by U.S. NIH R01 DC016875, R01 DC019404, UG3TR002636, U24HG010423, UH3TR002636, Curing Kids Massachusetts Eye & Ear and Ines-Fredrick Yeatts Fund (to Z.-Y.C.), NIH R01 DC019404 (to X.L. and Z.-Y.C.), and Margaret Q. Landenberger Research Foundation (to B.P.K.). Also, by U.S. NIH UG3AI150551, U01AI142756, R35GM118062, RM1HG009490, and HHMI (to D.R.L). R01DC012115, R01DC005575, and DOD RH220053 (to X.L.). The National Natural Science Foundation of China (NSFC882122019), Program for Professors with Special Appointments (Eastern Scholar) at Shanghai Institutions of Higher Learning (to Y.T.). We thank the Harvard Medical School Electron Microscopy Facility for their help on the Scanning Electron Microscopy Imaging.

## Author contributions

Conceptualization: Y.T., V.L., J.Z., D.T., D.R.L., and Z.Y.C. Methodology: Y.T., V.L., W.D., Y.L., M.N.W., W.Z., J.Z., D.T., Y.S., X.G., J.H., C.P., B.P.K., D.R.L., and Z.Y.C. Experiment: V.L., Y.T., W.D., Y.L., M.N.W., W.Z., J.Z., D.T., Y.S., X.G., J.H., A.P.R., and C.P. Data analysis: all authors. Supervision: D.R.L. and Z.Y.C. Manuscript writing: V.L., Y.L., Y.T., and Z.Y.C. Manuscript review and editing: Y.T., V.L., W.D., Y.L., M.N.W., W.Z., J.Z., D.T., Y.S., X.G., J.H., C.P., B.P.K., D.R.L., Z.Y.C., W.J.K., X.L., and H.W.

## Competing interests

Z.-Y.C. is a cofounder of Salubritas Therapeutics. He and D.R.L. have filed patent applications on: Efficient delivery of therapeutic molecules in vitro and in vivo" (15/523325) and "Method for efficient delivery of therapeutic molecules in vitro and in vivo" (15/523321). B.P.K. is an inventor on patents and patent applications filed by Mass General Brigham that describe genome engineering technologies. B.P.K. is a consultant for Avectas Inc., EcoR1 capital, and ElevateBio, and is an advisor to Acrigen Biosciences and Life Edit Therapeutics. X.L. is a SAB member of Rescue Hearing Inc, and a SAB member of Salubritas Therapeutics. The other authors declare no competing interests.
