## [Peer Review File · Nature Communications]

Treatment of monogenic and digenic dominant genetic hearing loss by CRISPR-Cas9 ribonucleoprotein delivery in vivoREVIEWER COMMENTS

Reviewer #1 (Remarks to the Author):

The authors deploy liposome-mediated in vivo delivery of CRISPR-Cas9 RNP complexes to edit a dominant hearing loss mutation in the *Atp2b2* Obl/+ mouse in an effort to restore auditory function. DPOAE and ABR data sets demonstrate significant reduction in auditory thresholds that support impressive startle responses at 8 weeks post-inoculation. The data are strong and the result compelling. However, there are several concerns that detract from the impact of the paper. ABR and DPOAE thresholds at 8 and 16 weeks after therapeutic inoculation progressively elevate and this phenotypic outcome in the *Atp2b2* Obl/+ mice is not clearly articulated. The lability of the therapeutic outcome should be articulated and discussed openly to begin to define improvements in the approach. The use of phalloidin to define stereociliary bundle phenotypes in treated mutants is far less impactful than SEM which would have been a far more compelling analysis. Critically, the unqualified assertion that hearing was rescued in the di-genic mouse model is not supported by the data: no DPOAE threshold improvement was achieved and only exceedingly modest reduction in low frequency ABR thresholds was detected that may not reflect clinically relevant hearing restoration. Major and minor concerns are detailed below.

Major Concerns

1) L324: "Significant hearing rescue in the *Atp2b2*Obl/+ mice treated with Cas9/*Atp2b2*-mut1/Lipo2000 was maintained at eight and 16 weeks after injection." The authors are commended for quantifying hearing at 8 and 16 weeks. However, the results are written as though hearing improvements are substantially preserved through 16 weeks, when in fact there is a progressive deterioration of hearing rescue at 8 and 16 weeks compared to the initial hearing sensitivity at 4 weeks. It's critically important to state this phenotypic lability in hearing rescue: sensitive DPOAE and ABR thresholds are progressively lost through 16 weeks post-treatment. Then, this instability in hearing rescue should be aggressively discussed in the context of the Cas9 editing strategy to offer possible explanations and paths forward for improvement. The auditory rescue at 4 weeks is indeed significant and impressive. The decline of DPOAE and ABR thresholds by 16 weeks does not detract from the findings of the study and must be vetted fully if productive experimental change is to be achieved.

2) L33 abstract: "We further show in a double-dominant mutant mouse model, in which the *Tmc1* Beethoven mutation and the *Atp2b2* Oblivion mutation cause di-genic genetic HL, RNP delivery targeting both mutations lead to hearing rescue." This statement must be removed from the abstract since it is not supported by the data. No DPOAE threshold restoration was observed in the treated digenic mice indicating that productive editing in outer hair cells was not achieved. ABR thresholds of between 75- and 80-dB SPL at 5-16 kHz in treated digenic mice are still exceedingly high compared to wild type controls and likely do not represent clinically relevant hearing. The authors would need to show persistence of the ABR responses detected at 4 weeks through 8 weeks, also with a dramatically improved startle response at 8 weeks, to assert that even limited hearing rescue was achieved. The

assertion in the abstract quoted above is not supported by the digenic mouse data presented and must be removed. In fact, the data from digenic mouse experiments indicate that dual gene Cas9 therapy will require more work to advance successfully, and consideration should be given to removing these data from the manuscript and focusing on Atp2b2 alone.

3) L493: Again, to assert the hearing was rescued significantly when no DPOAE thresholds were restored and ABR thresholds between 75- and 80-dB SPL from 5-16 kHz is not acceptable. The assertion detracts from the impact of this study and should be omitted.

4) The use of phalloidin to define the stereociliary bundle phenotype in treated and untreated mice is less impactful than scanning electron microscopy which is a standard strategy for such an analysis since a larger swath of the sensory epithelium could be shown in higher resolution. This is a missed opportunity that detracts from the strength of the biological effects produced by editing. Strong consideration must be given to conducting SEM on treated and control cochlea to heighten the impact of the data set.

5) In Figure 3D, include Wave I amplitudes of the control C3H uninjected mice so that the reader can make this essential comparison.

6) The authors should discuss whether the liposome-mediated gene editing strategy is expected to be efficacious after inoculation in mature, hearing ears versus the immature P0-P2 inner ear. The P0-P2 mouse inner ear has limited regenerative capacity through about P8, and this may reflect genetic and epigenetic plasticity that uniquely supports gene editing. Have the authors attempted to inoculate four week old Atp2b2 inner ears to assess efficacy of the approach in an ear that more fully models the hearing human ear?

Minor Concerns

Introduction

1) L58: "The advent of genome editing has made possible to therapeutically intervene genetic diseases fundamentally at genomic DNA level." Made it possible? Revise for clarity.

Results

7) Change Atp2b2 mut DNA label in 1b to Atp2b2 obl DNA to match the graph in 1c.

8) L889: Legend to Figure 1: "The list of indels is not comprehensive, only the top representative 890 indels were shown." Change to are shown.

9) L154: Correct the space after the hyphen: "The design of genome editing- based therapies requires the assessment of potential off- target effects to determine the specificity of editing. . ."

10) L164: "We measured the indel frequency at each off-target site by NGS analysis in Cas9:Atp2b2-mut1 treated Obl-OC1 cells following nucleofection of RNP or plasmids DNA." Consider revision to "plasmid DNA" here and in L20 (legend to Supplementary Figure 2).

11) L167: Consider the following edit for clarity: "In contrast, RNP treatment exhibited no detectable editing events in either WT allele or other off-target sites (Supplementary Fig. 2c), consistent with the

editing in the OC1 cell lines (Supplementary Fig. 1c) and our previous findings that RNP delivery greatly reduces editing at off-target sites (8, 13).”

12) L194: “We observed an indel rate of 1% in the Obl Atp2b2 locus in the injected Atp2b2Obl/+ organ of Corti samples (Fig. 2a).” But Fig. 2a X axis description is Injected Atp2b2 obl/obl mice. Which is correct, the figure or the legend and results section? Please resolve this issue.

13) L228 and L229: “In uninjected Atp2b2Obl/+ hair cells. . .” and “In injected hair cells, an Obl:WT. . .” These statements are imprecise since RNP injected inner ears were experimentally generated, not RNP injected hair cells. Revise for accuracy.

14) L233: Explain precisely how the disruption of Obl in 25% of hair cells was estimated.

15) L237: The use of “Lipo2000:Cas9/sgRNA Atp2b2-mut1” changes the syntax of this phrase from what was previously introduced. Revert to the original format for clarity.

16) Figure 3d Legend: correct “2mon” to 2 mon for the blue Atp2 Obl/+ entry in the key.

17) Figure 3d Legend. The gray box with green is exceedingly difficult to read. Revise this representation. Define the scale bars.

18) L295: “Uninjected Atp2b2Obl/+ inner ears showed elevations in cochlear neural responses at 4-week of age.” Revise to 4-weeks of age.

19) Figure 4 title (L942) asserts that the treatment rescues hearing. In fact, the DPOAE and ABR rescue, while dramatic, is still not wild type hearing sensitivity. The figure title should be qualified to accurately represent the data generated.

20) L320: Edit to “at frequencies ranging from” instead of ranged.

21) L333: “From the in vitro assay, the gRNA Atp2b2-mut1 showed the highest editing discriminating the Obl allele. . .” Awkward sentence. Revise for clarity.

22) L60: change “ars” to ears.

Discussion

23) L392: “In mice, the Obl mutation is shown disrupt the resting activity of the Ca²⁺ pump. . .” Revise to “shown to”.

24) L402: “Further DPOAE thresholds in the treated inner ears were elevated comparing to uninjected control ears”. Consider “Furthermore,” and “compared to” for clarity.

25) L408: The rescue achieved is partial hearing rescue since ABR and DPOAE thresholds are not wild type thresholds. Accurately stating the degree of hearing restoration does not detract from the impact of the study. Revise the sentence accordingly.

26) L433: Again, explain how the 25% value was derived.

27) L507: The ABR and DPOAE data do not reflect moderate hearing improvement. There is no improvement DPOAEs, so the editing of Atp2b2 in outer hair cells was not effective in the dual sgRNA context. This assertion detracts from the impact of this study and should be omitted.

28) L515: The data provided are inadequate to support the statement of “rescue effect in the digenic mouse mutants.” The data on the digenic mouse rescue is not compelling and detracts from an otherwise significant study.

Reviewer #2 (Remarks to the Author):

In this paper Tao and coworkers use liposomal delivery to induce a therapeutic allele-specific knockout in the Oblivion mouse model. This model represents a hearing loss model originating from outer hair cells (OHCs). OHCs – in contrast to IHCs - do not connect to the brain, but perform amplification of the sound signal. They are essential for normal hearing. Most hair-cell specific genes are expressed in both IHCs and OHCs, thus the Oblivion model is a unique dominant hearing loss model, where the functional consequence of OHC targeting can be evaluated.

Overall the study is well designed and the functional results are clear. Liposome-mediated ribonucleoprotein (RNP) delivery led to efficient, sustained rescue of hearing in the models tested.

The study however needs more thorough molecular biology characterization, especially because the authors observe a large deletion. Although it has been shown that CRISPR can lead to large genomic rearrangements, deletion of an exon (exon 24 of the *Atp2b2* gene in this case) is unusual, particularly that nothing explains the two additional intronic cuts (i.e. they are not found in the off-target screen). For example, it is very important to analyze if the WT allele was affected by the large deletion (as the proposed intronic cuts are present in the WT). This is not trivial to investigate, as the mutation site is deleted (with the exon), hence it is unknown whether the read was a mutant or WT read. The following experiments should clarify the source and effect of the large deletion – these must be conducted before considering the acceptance of the manuscript:

1. NGS on the deleted reads to characterize heterogeneity
2. Analysis of the intronic sites for small indel formation (-997-1010 and +661-821 cut sites, on Fig. 2b). The authors should do 150 bp paired end sequencing which covers the Oblivion site from one end and the intronic sites from the other end – so that it can be evaluated whether the WT and/or the mutated reads are affected by the intronic cuts. The intronic cuts can happen simultaneously or independent of the target cut site.
3. mRNA level must be analyzed. This will be decisive whether CRISPR effects the mutant or WT alleles. Given *Atp2b2* is expressed only in OHCs, the mRNA level should be the most relevant to analyze as it reflects editing in the target cells. As we expect non-sense mediated decay of the edited RNA, there

should be a difference in read count for mutant vs. WT reads. Indels can also be apparent at the mRNA level.

4. Important is to see if in the WT animals CRISPR would affect splicing (due to intronic cut or deletion etc.). A simple PCR on the cDNA, using primers in exons 23 and 25 would answer this question.

5. Alternatively perform ddPCR on the genomic DNA (if the assay can discriminate WT vs mutant alleles)

6. The best option would be to perform UDiTaS sequencing.

Some of these points are further discussed below.

Major comments:

- On Fig 1e the indel profile is very unusual. First, the indels concentrate around 9 bp upstream from the PAM site. Indels are almost all the time expected 3-4bp upstream of the PAM as this is the site where SpCas9 would cut. Likely the PAM site in red is wrongly annotated on the Figure – it should be the TGG more the left (upstream). Actually the mut1 gRNA indeed targets the more proximal TGG site, if Fig. 1a is correct. Please correct the PAM annotation.

- Second, the Fig 1e indel profile of the heterozygous cells shows that the unedited WT allele (TCC) is only present in 39% of the reads, whereas the unedited mutant allele (TTC) is present in 57% of the reads. Whether the deleted reads come from mutant or WT reads, cannot be identified as the mutated nucleotide itself is deleted. However it is unusual to see that there is a decrease of WT reads %, it should be that the unedited mutated reads have lower abundance. How do the authors explain the skew in WT/mut read distribution despite very low (less than 3%) indel percentage? This should not be the case. Can it be that there are large deletions, similarly to the in vivo experiments?

- The large deletions in vivo are very surprising, given a single cut. Also on Fig. 2A (NGS trace) there is a 10% difference between mutant and WT reads, again WT being less abundant. Is this reproducible? It would be useful to repeat the experiments a few times and plot the WT/mut read abundancies (and also in the case of control, to see if there is a PCR bias) and attempt to perform ddPCR to quantify the number of WT and mutant reads with and without CRISPR, in order to avoid confounding from PCR bias.

- Fig. 2e (sorted cells) seems to contradict Fig 2a and Fig 1e, where WT allele was less abundant.

- Although the authors claim in the text they performed NGS on the PCR product with deletion (line 209), the Figure legend states 'Sanger sequencing', and no NGS is shown. Perform the NGS experiment.

Minor comments:

- Suppl. Fig 1 – highlight the mutation site on NGS reads. Also, the read lengths should be the same size – it is unclear why the read length is different between mut 1, mut 2 and mut 3. Resolution of the Figure is low
- Fig 3d – it is difficult to appreciate the difference in stereocilia morphology. Higher magnification images or scanning EM would be more appropriate here.
- Not clear sentence on line 362: : we bred two dominant mouse models *Atp2b2*Obl/+ with *Tmc1*Bth/+ (delete 'two')
- I recommend to create a main Figure for the digenic approach (show gRNAs, one-two panel of important rescue data and sequencing)
- Line 392 – 394 – describe the mechanism of the mutation a bit better – difficult to understand

REVIEWER COMMENTS

Reviewer #1 (Remarks to the Author):

The authors deploy liposome-mediated in vivo delivery of CRISPR-Cas9 RNP complexes to edit a dominant hearing loss mutation in the *Atp2b2* *Obl/+* mouse in an effort to restore auditory function. DPOAE and ABR data sets demonstrate significant reduction in auditory thresholds that support impressive startle responses at 8 weeks post-inoculation. The data are strong and the result compelling. However, there are several concerns that detract from the impact of the paper. ABR and DPOAE thresholds at 8 and 16 weeks after therapeutic inoculation progressively elevate and this phenotypic outcome in the *Atp2b2* *Obl/+* mice is not clearly articulated. The lability of the therapeutic outcome should be articulated and discussed openly to begin to define improvements in the approach. The use of phalloidin to define stereociliary bundle phenotypes in treated mutants is far less impactful than SEM which would have been a far more compelling analysis. Critically, the unqualified assertion that hearing was rescued in the di-genic mouse model is not supported by the data: no DPOAE threshold improvement was achieved and only exceedingly modest reduction in low frequency ABR thresholds was detected that may not reflect clinically relevant hearing restoration. Major and minor concerns are detailed below.

Response: We thank the reviewer for raising the important points. Indeed, we observed robust hearing rescue in the injected ears one month later. The effect starts to lessen over time. By 4 months, significant hearing rescue persisted but was diminished compared to one month result. We have discussed extensively on the topic (see detailed response to question 1). We are in full agreement with the reviewer that the understanding of the issue will be significant in the future development for the clinic.

We have performed additional injection in the *Atp2b2*^{*Obl/+*} mice and conducted SEM study. We obtained the data to show the rescue of hair cell stereocilia morphology in the injected ears compared to the uninjected ears. A full response is provided to question 4.

To address the question raised regarding the treatment in the digenic model, we again performed additional injections with the data included in the analysis. First, the new data obtained is consistent with the data from previous injections by lower ABR and DPOAE thresholds. Second, when analyzed together, we not only detected significantly lower ABR thresholds in 5 out of 7 frequencies, but also a significantly lower DPOAE threshold at 16 kHz. Our data thus demonstrate hearing rescue in the digenic model by editing. The extent of rescue is less effective than that from editing in the single mutation model which we have discussed extensively in the manuscript. The feasibility of editing more than one mutations by a single injection will become increasingly important as more deaf patients are identified to have digenic mutations. We have provided detailed response to questions 2 and 3 below.

Major Concerns

1) L324: "Significant hearing rescue in the *Atp2b2**Obl/+* mice treated with Cas9/*Atp2b2*-

mut1/Lipo2000 was maintained at eight and 16 weeks after injection.” The authors are commended for quantifying hearing at 8 and 16 weeks. However, the results are written as though hearing improvements are substantially preserved through 16 weeks, when in fact there is a progressive deterioration of hearing rescue at 8 and 16 weeks compared to the initial hearing sensitivity at 4 weeks. It’s critically important to state this phenotypic lability in hearing rescue: sensitive DPOAE and ABR thresholds are progressively lost through 16 weeks post-treatment. Then, this instability in hearing rescue should be aggressively discussed in the context of the Cas9 editing strategy to offer possible explanations and paths forward for improvement. The auditory rescue at 4 weeks is indeed significant and impressive. The decline of DPOAE and ABR thresholds by 16 weeks does not detract from the findings of the study and must be vetted fully if productive experimental change is to be achieved.

Response: We thank the reviewer for the comments and valuable suggestions for improvement. In virtually all hearing rescue studies (including gene therapy and genome editing), the treatment effect starts to diminish over time. Thus, the maintenance of the rescue effect is of critical importance in the development of therapy for the clinic. While the mechanism for the reduced effect over time in our study is unknown, there are a few possible explanations. One hypothesis is that the duration of hearing rescue is related to the equilibrium between edited and unedited hair cells. In our study, we estimated ~20% of hair cells were edited, leaving more hair cells unedited. As a result, if unedited hair cells undergo degeneration, the edited hair cells may not be sufficient to maintain the health and function of the cochlea over time, leading to diminished cochlear function. To improve the rescue duration, further improvement in editing efficiency so more hair cells are edited will likely be critical. Another explanation is that the hair cells may have started to degenerate at the time of injection (p0-p2) whereas editing delays the degeneration process. Indeed, significant elevation of DPOAE and ABR was seen at one month of age, suggesting hair cells may have undergone some degeneration at an early stage. In this case, intervention at an even earlier age (e.g. in the embryos) may improve the rescue effect. The significant difference between early and later interventions has been abundantly illustrated in most hearing rescue studies in which the treatment was only effective when the injection was done in the neonatal age, whereas it would fail if the intervention were done a few days later (e.g. p6). We have rewritten the result and added discussion about the important topic. Please see page 13, lines: 374-376; and page 16, lines: 463-481.

2) L33 abstract: “We further show in a double-dominant mutant mouse model, in which the Tmc1 Beethoven mutation and the Atp2b2 Oblivion mutation cause di-genic genetic HL, RNP delivery targeting both mutations lead to hearing rescue.” This statement must be removed from the abstract since it is not supported by the data. No DPOAE threshold restoration was observed in the treated digenic mice indicating that productive editing in outer hair cells was not achieved. ABR thresholds of between 75- and 80-dB SPL at 5-16 kHz in treated digenic mice are still exceedingly high compared to wild type controls and likely do not represent clinically relevant hearing. The authors would need to show

persistence of the ABR responses detected at 4 weeks through 8 weeks, also with a dramatically improved startle response at 8 weeks, to assert that even limited hearing rescue was achieved. The assertion in the abstract quoted above is not supported by the digenic mouse data presented and must be removed. In fact, the data from digenic mouse experiments indicate that dual gene Cas9 therapy will require more work to advance successfully, and consideration should be given to removing these data from the manuscript and focusing on *Atp2b2* alone.

Response: We thank the reviewer for raising an important point. We have performed an additional study targeting both the *Bth* mutation in the *Tmc1* and the *Obl* mutation in the *Atp2b2* using the digenic mouse model. When all the data were analyzed, we confirmed the rescue in ABR and showed that the rescue in DPOAE was significant at 16 kHz (see Fig 6). The ability to target two mutations by one injection has important implications as more digenic mutations have been discovered in hearing loss patients and future treatment will require intervention in more than one mutation at the same time. Our data, while less robust than editing a single mutation, can be explained in part by the constraint in the delivery. It is well-known that editing efficiency is correlated with the level of editor and sgRNAs. We could only deliver 1µl of the RNP complex into one inner ear and have to reduce the sgRNA by one-half targeting each mutation, which could significantly reduce editing efficiency. Future studies testing high dose sgRNAs, as well as the modifications that improve sgRNA stability, will likely improve the treatment outcome. In humans, it is estimated that as much as 50-100 µl can be injected per cochlea, which should allow the injection of the mixture of two gRNAs with a sufficient dose targeting two mutations. We have extensively revised the section and pointed out a partial hearing rescue in the digenic model. Please see Abstract on page 2; page 14, lines 419-426; page 19, lines 568-572; and new Fig.6.

3) L493: Again, to assert the hearing was rescued significantly when no DPOAE thresholds were restored and ABR thresholds between 75- and 80-dB SPL from 5-16 kHz is not acceptable. The assertion detracts from the impact of this study and should be omitted.

Response: We appreciate the point of view raised by the reviewer. Our data showed conclusively partial hearing rescue by significant reductions in the ABR thresholds at five frequency regions by an average of 10 dB (an average of 14 dB reduction at four frequency regions) by editing both mutations. Further, we have shown a significant reduction in the DPOAE threshold at 16 kHz (please see Fig.6). We have provided a rationale for why we think the part is important to the current study. As a comparison, in our first editing study of the *Tmc1 Bth* mouse model (Gao., et al Nature 2018), the average ABR threshold reduction was ~17 dB even though it only targeted one mutation. Our goal is to use the current study as the baseline, so we and others can improve upon the rescue in future studies,

4) The use of phalloidin to define the stereociliary bundle phenotype in treated and untreated mice is less impactful than scanning electron microscopy which is a standard strategy for such an analysis since a larger swath of the sensory epithelium could be shown in higher

resolution. This is a missed opportunity that detracts from the strength of the biological effects produced by editing. Strong consideration must be given to conducting SEM on treated and control cochlea to heighten the impact of the data set.

Response: We thank the reviewer for the suggestion. We have performed additional injections and used scanning electron microscopy to examine hair cell stereocilia structure. We provided the evidence that at 4 weeks, the unedited hair cells show the sign of degeneration including the missing shorter stereocilia in the outer hair cells and the disorganization of the stereocilia of the inner hair cells, whereas the edited hair cells show relatively normal morphology. Please see page 11, lines: 298-309, and new Fig.4.

5) In Figure 3D, include Wave I amplitudes of the control C3H uninjected mice so that the reader can make this essential comparison.

Response: We thank the reviewer for the suggestion. We re-made the figure by adding wave 1 amplitudes of the control C3H uninjected mice data in the new Fig.5d.

6) The authors should discuss whether the liposome-mediated gene editing strategy is expected to be efficacious after inoculation in mature, hearing ears versus the immature P0-P2 inner ear. The P0-P2 mouse inner ear has limited regenerative capacity through about P8, and this may reflect genetic and epigenetic plasticity that uniquely supports gene editing. Have the authors attempted to inoculate four week old *Atp2b2* inner ears to assess efficacy of the approach in an ear that more fully models the hearing human ear?

Response: We thank the reviewer for the suggestion. Significant hearing loss is apparent at one month of age in the *Atp2b2^{obl/+}* mice. Our experience showed only intervention before the hearing loss can be successful in this model. That is why we did not try rescue at a later age as it is unlikely to work. We will examine in future studies the latest time point when the intervention is still efficacious.

Minor Concerns

Introduction

1) L58: "The advent of genome editing has made possible to therapeutically intervene genetic diseases fundamentally at genomic DNA level." Made it possible? Revise for clarity.

Response: We are sorry for the grammatical error. The sentence has been changed to " The advent of genome editing has made it possible to treat genetic diseases by the modifications at genomic DNA level to disrupt or correct the mutations", see page 4, lines: 74-75.

Results

7) Change *Atp2b2* mut DNA label in 1b to *Atp2b2 obl* DNA to match the graph in 1c.

Response: We thank the reviewer for noticing the inconsistency. We changed “mut” to “obl” accordingly (Fig.1b).

8) L889: Legend to Figure 1: “The list of indels is not comprehensive, only the top representative 890 indels were shown.” Change to are shown.

Response: We changed “were” to “are”.

9) L154: Correct the space after the hyphen: “The design of genome editing- based therapies requires the assessment of potential off- target effects to determine the specificity of editing. . .”

Response: We thank the reviewer for pointing the error. We corrected “editing- based” to “editing-based”.

10) L164: “We measured the indel frequency at each off-target site by NGS analysis in Cas9:Atp2b2-mut1 treated Obl-OC1 cells following nucleofection of RNP or plasmids DNA.” Consider revision to “plasmid DNA” here and in L20 (legend to Supplementary Figure 2).

Response: We thank the reviewer for pointing the error. We changed “plasmids” to “plasmid”.

11) L167: Consider the following edit for clarity: “In contrast, RNP treatment exhibited no detectable editing events in either WT allele or other off-target sites (Supplementary Fig. 2c), consistent with the editing in the OC1 cell lines (Supplementary Fig. 1c) and our previous findings that RNP delivery greatly reduces editing at off-target sites (8, 13).”

Response: We thank the reviewer for the suggestion. We edited this sentence as suggested.

12) L194: “We observed an indel rate of 1% in the Obl Atp2b2 locus in the injected Atp2b2Obl/+ organ of Corti samples (Fig. 2a).” But Fig. 2a X axis description is Injected Atp2b2 obl/obl mice. Which is correct, the figure or the legend and results section? Please resolve this issue.

Response: We thank the reviewer for pointing out the inconsistency. The description should be “Injected Atp2b2 obl/+ mice”. We have corrected the misstatement.

13) L228 and L229: “In uninjected Atp2b2Obl/+ hair cells. . .” and “In injected hair cells, an Obl:WT. . .” These statements are imprecise since RNP injected inner ears were experimentally generated, not RNP injected hair cells. Revise for accuracy.

Response: We have changed “injected hair cells” to “injected inner ears”.

14) L233: Explain precisely how the disruption of *Obl* in 25% of hair cells was estimated.

Response: The *Obl*:WT ratio percentages for uninjected samples are 52.57 ± 3.47 : 47.5 ± 3.48 (mean \pm SD), while those for injected samples are 46.18 ± 2.55 : 53.25 ± 2.27 (mean \pm SD). The resulting *Obl*/WT ratios are 1.11 ± 0.07 for uninjected samples and 0.87 ± 0.05 for injected samples. After normalization, the data shows a decrease in *Obl* % by $21.64 \pm 4.4\%$, indicating an editing efficiency of $21.64 \pm 4.4\%$. We thus used the estimated 21% in the manuscript. Please see page 9, lines 254-256.

In addition, we have performed a new study to determine the mRNA level of the *Obl* and the WT alleles after editing. The result showed downregulation of ~20% of the *Obl* transcripts (Fig.2i,j), which matched the editing frequency calculated from the NGS data of the DNA. The new data is presented in Fig. 2i,j and in the text pages 9-10, lines:256-276.

15) L237: The use of “Lipo2000:Cas9/sgRNA *Atp2b2*-mut1” changes the syntax of this phrase from what was previously introduced. Revert to the original format for clarity.

Response: We are grateful for the reviewer’s suggestion. We made the change accordingly.

16) Figure 3d Legend: correct “2mon” to 2 mon for the blue *Atp2 Obl*/+ entry in the key.

Response: We have made the correction.

17) Figure 3d Legend. The gray box with green is exceedingly difficult to read. Revise this representation. Define the scale bars.

Response: We thank the reviewer for the suggestion. We have changed to Fig. 3d. In addition, the new SEM images provide solid evidence of the difference in stereocilia between editing and unedited hair cells; please see new Fig. 4.

18) L295: “Uninjected *Atp2b2Obl*/+ inner ears showed elevations in cochlear neural responses at 4-week of age.” Revise to 4-weeks of age.

Response: We thank the reviewer for the suggestion. We have corrected the sentence accordingly.

19) Figure 4 title (L942) asserts that the treatment rescues hearing. In fact, the DPOAE and ABR rescue, while dramatic, is still not wild type hearing sensitivity. The figure title should be qualified to accurately represent the data generated.

Response: We thank the reviewer for the suggestion. We believe the title is an accurate description of the study in which we developed a treatment that rescues hearing in the

mouse model. The rescue is not complete, which has been extensively discussed in the manuscript. Please see page 16, lines: 463-481.

20) L320: Edit to “at frequencies ranging from” instead of ranged.

Response: We have made the change as suggested.

21) L333: “From the in vitro assay, the gRNA Atp2b2-mut1 showed the highest editing discriminating the Obl allele. . .” Awkward sentence. Revise for clarity.

Response: We thank the reviewer for the suggestion. We rewrote the sentence.

22) L60: change “ars” to ears.

Response: We are sorry for the error. The mistake has been corrected.

Discussion

23) L392: “In mice, the Obl mutation is shown disrupt the resting activity of the Ca²⁺ pump. . .” Revise to “shown to”.

Response: We have corrected the error.

24) L402: “Further DPOAE thresholds in the treated inner ears were elevated comparing to uninjected control ears”. Consider “Furthermore,” and “compared to” for clarity.

Response: We thank the reviewer for the suggestion. We have made the change as suggested.

25) L408: The rescue achieved is partial hearing rescue since ABR and DPOAE thresholds are not wild type thresholds. Accurately stating the degree of hearing restoration does not detract from the impact of the study. Revise the sentence accordingly.

Response: We agree with the reviewer’s comments. We have revised the sentence to describe the partial hearing rescue by editing more accurately.

26) L433: Again, explain how the 25% value was derived.

Response: The estimated editing efficiency is ~21%. Please see the reply to the comment 14) from the reviewer 1.

27) L507: The ABR and DPOAE data do not reflect moderate hearing improvement. There is no improvement DPOAEs, so the editing of Atp2b2 in outer hair cells was not effective in the

dual sgRNA context. This assertion detracts from the impact of this study and should be omitted.

Response: We appreciate the point raised by the reviewer. We performed additional injection study and our data showed conclusively hearing rescue by significant reductions in the ABR thresholds at five frequencies by an average of 10 dB. Further, we have shown a significant reduction in DPOAE threshold at 16 kHz. Please see page 14, lines: 419-426, and new Fig.6. The rescue effect in the digenic model is not as robust as in the single mutant which we offered possible explanations. Please see page 14, lines 419-426; page 19, lines 568-572;

28) L515: The data provided are inadequate to support the statement of “rescue effect in the digenic mouse mutants.” The data on the digenic mouse rescue is not compelling and detracts from an otherwise significant study.

Response: We have provided our rationale for why we believe our data support the conclusion of hearing rescue in the digenic mutant model, namely significant improvement in ABR and DPOAE threshold by editing. Please see page 14, lines 419-426; page 19, lines 568-572;

Reviewer #2 (Remarks to the Author):

In this paper Tao and coworkers use liposomal delivery to induce a therapeutic allele-specific knockout in the Oblivion mouse model. This model represents a hearing loss model originating from outer hair cells (OHCs). OHCs – in contrast to IHCs - do not connect to the brain, but perform amplification of the sound signal. They are essential for normal hearing. Most hair-cell specific genes are expressed in both IHCs and OHCs, thus the Oblivion model is a unique dominant hearing loss model, where the functional consequence of OHC targeting can be evaluated.

Overall the study is well designed and the functional results are clear. Liposome-mediated ribonucleoprotein (RNP) delivery led to efficient, sustained rescue of hearing in the models tested.

The study however needs more thorough molecular biology characterization, especially because the authors observe a large deletion. Although it has been shown that CRISPR can lead to large genomic rearrangements, deletion of an exon (exon 24 of the *Atp2b2* gene in this case) is unusual, particularly that nothing explains the two additional intronic cuts (i.e. they are not found in the off-target screen). For example, it is very important to analyze if the WT allele was affected by the large deletion (as the proposed intronic cuts are present in the WT). This is not trivial to investigate, as the mutation site is deleted (with the exon), hence it is unknown whether the read was a mutant or WT read. The following experiments should clarify the source and effect of the large deletion – these must be conducted before considering the acceptance of the manuscript:

Response: We thank the reviewer for the thoughtful suggestions. The identification of large deletions is a surprise to us, yet it explains the discrepancy between a relatively low indel rate in vivo and the robust hearing rescue. We have addressed the large deletions by new experiment to show that no small DNA fragments can be amplified by nested PCR using the DNA from injected WT mice. We analyzed the junction sequence in DNA by NGS from injected WT mice and did not detect any indels. We detected indels in the *Obl* cDNA from injected *Atp2b2*^{Obl/+} ears only. We have provided detailed responses to all the questions raised by the reviewer. Please see below.

1. NGS on the deleted reads to characterize heterogeneity

Response: We thank the reviewer for the suggestion for additional work to resolve the issue. We performed NGS study on the deletions. The process of in vivo editing efficiency evaluation is shown below. First, we used different sets of primers to amplify different length of products (~1.2 kb, ~1.4 kb, ~2.0 kb) in the *Atp2b2* locus. The PCR produced expected ~2 kb fragment but also unexpected smaller fragments in injected *Atp2b2*^{Obl/+} mice but not in injected WT mice. We performed NGS on the smaller fragments with the representative reads shown in Fig. 2d. So the production of the smaller fragments was associated with the mutant allele only. Since we could not amplify any small DNA fragment even with nested PCR using the DNA from injected WT mice, we therefore conclude that large deletions represented by small DNA fragments were the result of editing on the *Obl* allele. Please also see page 8, lines:215-239 for description.

2. Analysis of the intronic sites for small indel formation (-997-1010 and +661-821 cut sites, on Fig. 2b). The authors should do 150 bp paired end sequencing which covers the Oblivion site from one end and the intronic sites from the other end – so that it can be evaluated whether the WT and/or the mutated reads are affected by the intronic cuts. The intronic cuts can happen simultaneously or independent of the target cut site.

Response: We thank the reviewer for the suggestion. We performed an additional NGS study on the intronic sites for small indels from injected *Oblivion*/+ or +/+ cells using pairs of primers as suggested. We amplified the intronic sites and did not detect any small indels, which supports that the large deletions are the result of on-target cutting only. Please see the new Supplementary Fig 2f.

The observation of large deletions due to single on-target cut caused genetic lesions has been reported previously (M. Kosicki, et. al. Nat. Biotechnol. 2018). Please see the new Supplementary Fig 2f. and in manuscript page 9, lines: 231-239

3. mRNA level must be analyzed. This will be decisive whether CRISPR effects the mutant or WT alleles. Given *Atp2b2* is expressed only in OHCs, the mRNA level should be the most relevant to analyze as it reflects editing in the target cells. As we expect non-sense mediated decay of the edited RNA, there should be a difference in read count for mutant vs. WT reads. Indels can also be apparent at the mRNA level.

Response: We thank the reviewer for the suggestion. We have performed a new study by injecting Cas9/gRNA into the mutant inner ears and harvested the cochlea for cDNA synthesis of *Atp2b2*. The uninjected ears served as control. The NGS study showed small indels in the *Atp2b2 Oblivion* allele, demonstrating editing on the *Obl* allele. We compared the ratio between *Obl* and WT *Atp2b2* cDNAs, in the injected and uninjected control ears, and found an average reduction of 20% of the *Obl* transcripts in the injected ears. The mRNA study result closely matched the estimated editing efficiency from the comparison of the ratio between WT and *Obl* alleles. The new data is presented by Fig. 2i, j, and in the text page 9 lines: 254-266.

4. Important is to see if in the WT animals CRISPR would affect splicing (due to intronic cut

or deletion etc.). A simple PCR on the cDNA, using primers in exons 23 and 25 would answer this question.

Response: We performed an experiment to determine if there are exon 24 skipped transcripts using primers in exons 23 and 25. In WT animals, we did not detect any exon 24 skipped mRNA after injection, please see new Fig. 2h. This is consistent with our results showing that there is no large deletion or intronic cut in the WT allele by the editing complex. Please see p10, lines 268-276.

5. Alternatively perform ddPCR on the genomic DNA (if the assay can discriminate WT vs mutant alleles).

Response: We thank the reviewers for the recommendation. ddPCR is technically challenging, and we do not have the setup in the lab. Our additional experiment above has adequately addressed the concerns.

6. The best option would be to perform UDiTaS sequencing.

Response: We thank the reviewers for the recommendation. Again, UDiTaS sequencing is technically challenging, and we do not have the setup in the lab. Our additional experiment above has adequately addressed the concerns.

Some of these points are further discussed below.

Major comments:

- On Fig 1e the indel profile is very unusual. First, the indels concentrate around 9 bp upstream from the PAM site. Indels are almost all the time expected 3-4bp upstream of the PAM as this is the site where SpCas9 would cut. Likely the PAM site in red is wrongly annotated on the Figure – it should be the TGG more the left (upstream). Actually the mut1 gRNA indeed targets the more proximal TGG site, if Fig. 1a is correct. Please correct the PAM annotation.

Response: We thank the reviewer for pointing out the error. We performed an additional experiment and confirmed the cutting site as the reviewer indicated, and have re-made the figure (Fig. 1d, e) to replace the previous Fig. 1d, e.

• Second, the Fig 1e indel profile of the heterozygous cells shows that the unedited WT allele (TCC) is only present in 39% of the reads, whereas the unedited mutant allele (TTC) is present in 57% of the reads. Whether the deleted reads come from mutant or WT reads, cannot be identified as the mutated nucleotide itself is deleted. However it is unusual to see that there is a decrease of WT reads %, it should be that the unedited mutated reads have lower abundance. How do the authors explain the skew in WT/mut read distribution despite very low (less than 3%) indel percentage? This should not be the case. Can it be that there are large deletions, similarly to the in vivo experiments?

Response: We thank the reviewer for pointing out the discrepancy. We performed a new experiment with the data that showed 50% of the WT allele, 45% of the unedited mutant allele, and the rest being indels. The data is included in the new figure Fig. 1e which replaces the previous figure.

• The large deletions in vivo are very surprising, given a single cut. Also on Fig. 2A (NGS trace) there is a 10% difference between mutant and WT reads, again WT being less abundant. Is this reproducible? It would be useful to repeat the experiments a few times and plot the WT/mut read abundancies (and also in the case of control, to see if there is a PCR bias) and attempt to perform ddPCR to quantify the number of WT and mutant reads with and without CRISPR, in order to avoid confounding from PCR bias.

Response: We thank the reviewer for the question and suggestion. We made an extra effort to identify large deletions in order to understand the apparent discrepancy between robust hearing rescue and a modest indel rate. The detection of large deletions and the analysis of Obl:WT allele frequency using purified hair cells provided insight into the discrepancy. In fact, editing on the Obl site was efficient that also involved large deletions, which with indels collectively reduced the Obl frequency, leading to robust hearing rescue.

We have performed a new injection experiment as suggested and obtained data that showed the indels, which is now part of the new figure (Fig.2b, c). We also performed injection in WT mice, and collected the inner ear DNA for nest PCR. We did not detect any

smaller fragments in WT uninjected, WT injected, *Atp2b2*^{Obl/+} uninjected animals from multiple samples. Only in the injected *Atp2b2*^{Obl/+} group did we detect small fragments by nested PCR; please see the new figure (Fig. 2e). Together, the data demonstrated the on-target cutting at the mutant allele produced large deletions. Please find the description on page 8, line 215-239: The sequence data shown in Fig. 2f is representative reads of different NGS data from multiple injected samples, showing the large deletion. The study provides a method to develop a more comprehensive approach to studying editing effect beyond indel detection.

• Fig. 2e (sorted cells) seems to contradict Fig 2a and Fig 1e, where WT allele was less abundant.

Response suggestion: While we should see a ratio of 50% between WT and the mutant allele DNA in the heterozygous mice, from the NGS study of isolated hair cells, we have consistently seen a biased ratio between WT and the mutant, likely due to uneven amplification of DNAs as the result of very few cells collected (Gao, et al., Nature 2018). In this study, in the uninjected inner ear, the ratio of WT vs mutant is 45%:55%, which suggests the method preferentially amplified the mutant vs WT allele. In the injected ear, the ratio between WT vs. the mutant is 55%:45%. We have calculated the editing efficiency to be ~21% based on the data, which is consistent with the data on the reduction (~20%) of the

mutant transcripts. Please also see the reply to reviewer 1, (14) L233, for detailed explanation.

- Although the authors claim in the text they performed NGS on the PCR product with deletion (line 209), the Figure legend states ‘Sanger sequencing’, and no NGS is shown. Perform the NGS experiment.

Response: We thank the reviewer for the suggestion. We conducted the NGS analysis on the PCR products with deletions from multiple experiments, with the representative data shown in Fig. 2f.

Minor comments:

- Suppl. Fig 1 – highlight the mutation site on NGS reads. Also, the read lengths should be the same size – it is unclear why the read length is different between mut 1, mut 2 and mut 3. Resolution of the Figure is low

Response: We thank the reviewer for the suggestion. We made the changes to the Fig. s1 according to the suggestion

- Fig 3d – it is difficult to appreciate the difference in stereocilia morphology. Higher magnification images or scanning EM would be more appropriate here.

Response: We have performed additional in vivo injections and the SEM study to show the preservation of stereocilia integrity of the hair cells in the injected inner ear. The data is included in the new figure Fig.4.

We performed new scanning EM study with the data that are included in the new figure 4.

- Not clear sentence on line 362: we bred two dominant mouse models *Atp2b2*^{Obl/+} with *Tmc1*^{Bth/+} (delete ‘two’)

Response: We thank the reviewer for the suggestion. We made the change to the sentence: “we bred the dominant mouse model *Atp2b2*^{Obl/+} with another dominant model *Tmc1*^{Bth/+}...”

- I recommend to create a main Figure for the digenic approach (show gRNAs, one-two panel of important rescue data and sequencing)

Response: We thank the reviewer for the suggestion. We made a figure for the digenic approach that is part of the new main figure Fig.6.

- Line 392 – 394 – describe the mechanism of the mutation a bit better – difficult to understand

Response: We thank the reviewer for pointing out the issue. We have rewritten the sentence so it is easier to understand. Please find the sentence on page 15, lines 437-440.

REVIEWER COMMENTS

Reviewer #1 (Remarks to the Author):

The authors seek to validate a CRISPR/Cas9 strategy to treat monogenic and digenic forms of dominant hearing loss. Conceptually, the idea is to selectively disrupt the dominant Oblivion allele of *Atp2b2* and in the digenic model, disrupt the Oblivion allele along with the dominant Beethoven allele of *Tmc1* in the double mutant. The expectation is that after disruption of dominant mutant allele(s), the intact wild type allele will be sufficient to restore clinically-relevant auditory function. The revised manuscript is strengthened by the inclusion of the qualitative SEM data that describes an improvement in bundle morphology after therapeutic treatment. The overarching weakness is the assertion that the CRISPR-Cas9 strategy was effective at rescuing hearing in the digenic mouse model when the ABR and DPOAE data do not support this conclusion.

Primary concerns on the Obl allele

L311: “In vivo editing on the Obl mutation of *Atp2b2* gene restores OHC function and rescues hearing”
The “rescues hearing” in the section title implies that wild type thresholds were achieved, and while the responses are impressive, they are not wild type thresholds but would certainly benefit the animal in behaviorally significant ways. Leave “rescues hearing” for the person who devises a strategy that actually restores wild type ABR and DPOAE thresholds. There is statistically significant recovery of DPOAE thresholds and ABR thresholds that is highly likely to be behaviorally relevant and the title of the section should reflect this and not full recovery of hearing. Revise the section title.

L374: “From 4 weeks to 16 weeks post injection, while hearing rescue in the *Atp2b2*Obl/+ mice treated with Cas9/*Atp2b2*-mut1/Lipo2000 persisted, the reductions in DPOAE and ABR thresholds became progressively smaller.” This is a deeply encumbered statement that does not clearly summarize the data set. First, partial hearing recovery persisted and this must be stated clearly. Avoiding use of the term “rescue” altogether would strengthen the paper. Second, the ABR and DPOAE thresholds at 16 weeks would not effectively support species specific behaviors in the wild (i.e., finding food, avoiding predation, securing a mate, etc.): the mice are effectively behaviorally deaf by 16 weeks post treatment. That is the fascinating observation: the strategy partially rescued ABR and DPOAEs but then at a systems level the hearing falters over the next 12 weeks. The partial rescue at 4 weeks addresses conceptually the feasibility of the approach, and the lability of the rescue directs us to consider how to achieve sustained rescue. Presenting the data in a clear and unencumbered way will focus attention on defining relevant solutions. Supplementary Figure 3 must plot the DPOAE and ABR thresholds achieved at 4 weeks along with those at 8 and 16 weeks, and then present the statistical comparisons at 4, 8, and 16 weeks. Clearly the partial ABR and DPOAE rescue begins to falter at 8 weeks and is substantially compromised at 16 weeks. The results as written focus only on what hearing persists, but in doing so neglects the overarching effect which is the lability of the partial hearing restoration. This is a vital observation and it does not detract from the significance of the study to fully articulate these changes

clearly. Revise the figure so we can see the magnitude of the change in hearing clearly from 4 to 16 weeks.

L575: “Our work demonstrates the feasibility of RNP delivery of editing agents into the inner ear to target outer hair cell mutations and preserve hearing in the *Atp2b2Obl/+* mice.” Hearing was not preserved in the *ATP2b2 Obl/+* mice: ABR and DPOAE thresholds elevated dramatically by 16 weeks. The statement is not supported by the data.

Primary concerns on the digenic mouse model

L554: “It is significant that injection of Cas9 complexed with the double gRNAs of *Atp2b2-mut1* and *Tmc1-mut3* rescued hearing significantly whereas no hearing was rescued by a single gRNA delivery.” A statistically significant improvement in ABR thresholds at 5 frequencies was shown, yet the ABR thresholds are approximately 78-80 dB SPL. A statistically significant improvement in ABR thresholds does not translate into a clinically relevant functional outcome: these mice remain functionally deaf. Similarly, a statistically significant improvement in the DPOAE threshold at one frequency, 16 kHz, was shown, yet the DPOAE threshold was at least 75 dB SPL and is of absolutely no clinical relevance for the animal. The expectation for a gene therapy is that a clinically relevant degree of hearing is achieved. This expectation has not been met.

L568 “Despite the hearing rescue in the two-gRNA injected double mutant inner ears is not at the same level as each gRNA injected to *Atp2b2Obl/+* or *Tmc1Bth/+* individually, the partial hearing improvement observed after RNP delivery opens the possibility for further refinement of the approach as potential treatment of hearing loss of multigenic origin.” There was no clinically relevant degree of auditory recovery in the digenic mutant model. The data indicate that a different strategy must be used to achieve clinically relevant auditory recovery.

Minor

L295: Restructure the results to highlight the SEM data set which is a compelling way to qualitatively evaluate the bundle morphology. The phalloidin data are not compelling for analyzing bundle morphology. Figure 3d should be deleted, then lead in to Figure 4 SEM data. Revise the text in the results section accordingly.

L351 and Figure 5e: The average waveform comparison in Fig. 5e is used to conclude that in uninjected ears the ABR waveform pattern was absent. This statement is imprecise and confusing. ABR thresholds are defined at all frequencies tested in untreated mutants in Fig. 5b which requires a waveform pattern. Moreover, and a waveform pattern is present for the uninjected mutant in Fig. 5c. Regardless, rather

than rely on an average ABR waveform at one frequency and intensity, a metric of dubious value that is infrequently used, provide a supplementary table of pure tone thresholds for all experimental and control mice tested. Delete the average waveform cascade figure that does not support the conclusion asserted. The tabulated ABR pure tone threshold data would further the argument about the patency of the hearing rescue in a compelling way.

L469: ". . . the edited hair cells may not be sufficient to maintain the health and function of the cochlea over time, leading to diminished cochlear function." Is the argument that unedited hair cells die and this structurally or electrically destabilizes the rescued hair cells, negatively impacting their function? Be explicitly clear in your argument.

Supplementary Figure 4a,b: Since the *Atp2b2*-mut 2 allele targets both wild type and mutant alleles, wouldn't elevated ABR and DPOAE thresholds in treated *Obl*/+ animals be expected compared to controls? Why do you think this was not the case?

Abstract L51: ". . .RNP delivery targeting both mutations lead to partial hearing rescue." Leads to hearing rescue. Revise.

L88: "The auditory function requires functional inner and outer hair cells" Too many functions. Avoid the repetition of the same word.

L89: ". . . Further, multiple genetic hearing loss has been shown to be associated with mutations. . ." Clarify what is meant by multiple genetic hearing loss. Also, its associated with is better here. Revise.

L92: "OHC function shown by distortion product of acoustic emissions" Provide the correct articulation of DPOAE here.

L100: Provide *Obl* reference here.

L135: consider protospacer rather than spacer.

L137: Consider adding the cell line for the in vitro analyses here.

L909: Consider replacing the asterisks in 1b with the actual base pair sizes of the fragments. Same for Supplementary Figure 1a.

Figure 2a: Genomic DNA rather than Genome DNA. Revise.

L210: rate of 0.4~1.2%. Remove the approximately symbol and replace with a hyphen.

L275: instead of “in a significant number of hair cells, why not say in an estimated 21% of hair cells.

L343: State the frequency range over which the ABRs were conducted here.

Figure 5C: Define the gray boxes in the figure.

Reviewer #2 (Remarks to the Author):

The authors have performed several new experiments and the manuscript has improved.

I only have one comment for Figure 2e in the revised manuscript. On this Figure, the authors present new nested PCR analysis of the wild-type and mutant DNA, to investigate the presence of large deletions. The gel image is very inconsistent for the following reasons:

- 1) a different DNA ladder is used for the mutant allele and the wild-type allele.
- 2) for the WT, only one gel is shown, while for the mutant allele, two gels are shown that represent the two different PCRs.
- 3) In the uninjected mutant mice - the three animals show a different PCR result - the first animal has a smeary band, the second has 4 short bands and the last one has one band at around 100bp. It is not possible to conclude based on this PCR if the result from 3 different animals show 3 different profiles

4) The WT gel is cropped but short bands are half-visible. Are these primer dimers? If around 100bp, they are too long.

In general the PCR has to be optimised and the results should be presented side-by-side.

REVIEWER COMMENTS

Reviewer #1 (Remarks to the Author):

The authors seek to validate a CRISPR/Cas9 strategy to treat monogenic and digenic forms of dominant hearing loss. Conceptually, the idea is to selectively disrupt the dominant Oblivion allele of *Atp2b2* and in the digenic model, disrupt the Oblivion allele along with the dominant Beethoven allele of *Tmc1* in the double mutant. The expectation is that after disruption of dominant mutant allele(s), the intact wild type allele will be sufficient to restore clinically-relevant auditory function. The revised manuscript is strengthened by the inclusion of the qualitative SEM data that describes an improvement in bundle morphology after therapeutic treatment. The overarching weakness is the assertion that the CRISPR-Cas9 strategy was effective at rescuing hearing in the digenic mouse model when the ABR and DPOAE data do not support this conclusion.

Response: We are grateful for the reviewer's comments on the improvement of the revision based on the new data, and we have incorporated them into the revision. We believe our data support the feasibility of partial hearing recovery in the digenic model and will lay the foundation for further improvement.

First, we used the gold standard of ABR/DPOAE thresholds in the editing/gene therapy field to measure the rescue effect, which showed lower ABR/DPOAE thresholds that are statistically significant across 5 frequencies by ABR and one frequency by DPOAE. By the criteria, we demonstrated limited hearing recovery and established the feasibility of the approach applicable to digenic models. Second, the study is focused on the auditory function and not on the behaviors. We have changed the section title to **"Double editing partially recovers the auditory function of digenic mutation origin *in vivo*"**, p12, line 397. Third, we used a digenic model that is created artificially by a *Bth* and an *Obl* mutation. Since each mutation causes early onset (at one month) and severe hearing loss, the threshold to achieve any auditory function recovery in this model is high. In humans, if an intervention can be done in patients with, for example, a 70 dB hearing loss at a frequency, a 15 dB reduction in the ABR threshold could bring hearing to 55 dB, which is clinically relevant. Finally, we (or anyone else) do not know how the study can be translated into humans. This is one of the most pressing issues facing the whole field. Given that the first human clinical trial based on the mouse work has started and more to follow, we will soon learn the treatment efficacy, which should inform us about the translational potential.

Primary concerns on the *Obl* allele

L311: "In vivo editing on the *Obl* mutation of *Atp2b2* gene restores OHC function and rescues hearing" The "rescues hearing" in the section title implies that wild type thresholds were achieved, and while the responses are impressive, they

are not wild type thresholds but would certainly benefit the animal in behaviorally significant ways. Leave “rescues hearing” for the person who devises a strategy that actually restores wild type ABR and DPOAE thresholds. There is statistically significant recovery of DPOAE thresholds and ABR thresholds that is highly likely to be behaviorally relevant and the title of the section should reflect this and not full recovery of hearing. Revise the section title.

Response: We thank the reviewer for the comment. We accepted the reviewer’s suggestion and changed “hearing rescue” to “hearing recovery” throughout the text. We modified the section title to include “partially” (p9, line 302 and p11, line 352). In addition, we made the changes to the abstract to better reflect hearing recovery detected by “limited hearing recovery” (line 49 and “potentially” (line 52). For gene editing/gene therapy studies, ABR/DPOAE threshold changes with significance shown in statistical analysis are the standard measurements to evaluate the outcome. Our study uses the parameters as the readout for the auditory function, and we did not measure the behavioral changes. To our knowledge, among all the published studies using gene/editing for hearing loss, none of which could restore hearing to the WT level across all the frequencies long term.

L374: “From 4 weeks to 16 weeks post injection, while hearing rescue in the Atp2b2Obl/+ mice treated with Cas9/Atp2b2-mut1/Lipo2000 persisted, the reductions in DPOAE and ABR thresholds became progressively smaller.” This is a deeply encumbered statement that does not clearly summarize the data set. First, partial hearing recovery persisted and this must be stated clearly. Avoiding use of the term “rescue” altogether would strengthen the paper. Second, the ABR and DPOAE thresholds at 16 weeks would not effectively support species specific behaviors in the wild (i.e., finding food, avoiding predation, securing a mate, etc.): the mice are effectively behaviorally deaf by 16 weeks post treatment. That is the fascinating observation: the strategy partially rescued ABR and DPOAEs but then at a systems level the hearing falters over the next 12 weeks. The partial rescue at 4 weeks addresses conceptually the feasibility of the approach, and the lability of the rescue directs us to consider how to achieve sustained rescue. Presenting the data in a clear and unencumbered way will focus attention on defining relevant solutions. Supplementary Figure 3 must plot the DPOAE and ABR thresholds achieved at 4 weeks along with those at 8 and 16 weeks, and then present the statistical comparisons at 4, 8, and 16 weeks. Clearly the partial ABR and DPOAE rescue begins to falter at 8 weeks and is substantially compromised at 16 weeks. The results as written focus only on what hearing persists, but in doing so neglects the overarching effect which is the lability of the partial hearing restoration. This is a vital observation and it does not detract from the significance of the study to fully articulate these changes clearly. Revise the figure so we can see the

magnitude of the change in hearing clearly from 4 to 16 weeks.

Response: We thank the reviewer for the insight and suggestions. We have made changes to the text by describing the initial robust hearing preservation at 4 and 8 weeks with the effect that was reduced at 16 weeks. See p11, lines 357-369. We have re-made the SFig.3 to include the comparisons at 4, 8, and 16 weeks in ABR and DPOAE as suggested, which showed a similar reduction in the ABR/DPOAE thresholds from 4 to 8 weeks, with substantially higher ABR/DPOAR thresholds by 16 weeks compared to that of 4 and or 8 weeks.

The diminished treatment effect over time is one key area that needs to be focused on in the future, as it has also been observed in most gene and editing therapy studies for genetic hearing loss in mice. For our model, the potential reasons for the diminished treatment effect long term have been extensively discussed. Please see p14, lines 466-472. It is important to clarify that our study is focused on the auditory functions assessed by ABR/DPOAE, and we only examined the behavior as a result of hearing rescue by the startle reflex. We did not correlate with how the hearing recovery could affect other animal behaviors, as the reviewer mentioned.

L575: "Our work demonstrates the feasibility of RNP delivery of editing agents into the inner ear to target outer hair cell mutations and preserve hearing in the *Atp2b2*Obl/+ mice." Hearing was not preserved in the *ATP2b2* Obl/+ mice: ABR and DPOAE thresholds elevated dramatically by 16 weeks. The statement is not supported by the data.

Response: We thank the reviewer for the comment. We have changed the sentence to "Our work demonstrates the feasibility of RNP delivery of editing agents into the inner ear to target outer hair cell mutations and to recover hearing in the *Atp2b2*Obl/+ mice with the effect that is reduced long term.". Please see p17, lines 574-576.

Primary concerns on the digenic mouse model

L554: "It is significant that injection of Cas9 complexed with the double gRNAs of *Atp2b2*-mut1 and *Tmc1*-mut3 rescued hearing significantly whereas no hearing was rescued by a single gRNA delivery." A statistically significant improvement in ABR thresholds at 5 frequencies was shown, yet the ABR thresholds are approximately 78-80 dB SPL. A statistically significant improvement in ABR thresholds does not translate into a clinically relevant functional outcome: these mice remain functionally deaf. Similarly, a statistically significant improvement in the DPOAE threshold at one frequency, 16 kHz, was shown, yet the DPOAE threshold was at least 75 dB SPL and is of absolutely no clinical relevance for the animal. The expectation for a gene therapy is that

a clinically relevant degree of hearing is achieved. This expectation has not been met.

Response: We thank the reviewer for the comments and agree with the reviewer that we did not achieve a similar degree of hearing recovery in the digenic model as in the monogenic model. On the criteria of lowering ABR/DPOAE thresholds after injection, which is the gold standard used by the field, our data demonstrate the effect in the digenic model both by the statistical significance and the reduction of ~15 dB of ABR thresholds across many frequencies. DPOAE threshold lower threshold at one frequency that was statistically significant. We provide explanations as to why we detected a smaller recovery. Again, our work is **on auditory functions**, and it did not concern any behavioral changes. This study is focused on the animal model to produce proof-of-concept evidence to establish the feasibility so that we can continue the development to improve the efficacy. The digenic study here provides the information necessary for us to refine and improve the outcome. How the work from the mouse study can be translated into the clinic is a major issue for the field of editing/gene therapy for hearing loss in general. Ultimately the information can only come from clinical trials so that we can determine the type of hearing recovery in mice that can be translated to the patients.

We are very encouraged by the data, as our initial calculation based on editing two mutations simultaneously indicated that we would unlikely detect any recovery. The implication is that it's the first time that editing could be used to address two deafness mutations at the same time, given all the limitations. This result provides us and the community with evidence that we can further develop the option to address digenic hearing loss in the future.

L568 “Despite the hearing rescue in the two-gRNA injected double mutant inner ears is not at the same level as each gRNA injected to *Atp2b2*Obl/+ or *Tmc1*Bth/+ individually, the partial hearing improvement observed after RNP delivery opens the possibility for further refinement of the approach as potential treatment of hearing loss of multigenic origin.” There was no clinically relevant degree of auditory recovery in the digenic mutant model. The data indicate that a different strategy must be used to achieve clinically relevant auditory recovery.

Response: We thank the reviewer for the comment. We want to express the opinion that the hearing improvement in the digenic model, despite being moderate, opens a door for us to continue the path with further modifications by **the same RNP delivery strategy** (e.g., the modified sgRNA to improve the stability, injection by canalostomy with an increased volume, the use of more efficient editors and re-dosing) to improve overall outcome. The effect shown by editing to target two mutations at the same time illustrated a viable strategy in our view. This point is particularly relevant given the rapid advances in editing

technology and our data support that the improvement in editing efficiency will have the most drastic impact on the treatment outcome.

Minor

L295: Restructure the results to highlight the SEM data set which is a compelling way to qualitatively evaluate the bundle morphology. The phalloidin data are not compelling for analyzing bundle morphology. Figure 3d should be deleted, then lead in to Figure 4 SEM data. Revise the text in the results section accordingly.

Response: We thank the reviewer for the comment. We have deleted Figure 3d and revised the text in the results section accordingly.

L351 and Figure 5e: The average waveform comparison in Fig. 5e is used to conclude that in uninjected ears the ABR waveform pattern was absent. This statement is imprecise and confusing. ABR thresholds are defined at all frequencies tested in untreated mutants in Fig. 5b which requires a waveform pattern. Moreover, and a waveform pattern is present for the uninjected mutant in Fig. 5c. Regardless, rather than rely on an average ABR waveform at one frequency and intensity, a metric of dubious value that is infrequently used, provide a supplementary table of pure tone thresholds for all experimental and control mice tested. Delete the average waveform cascade figure that does not support the conclusion asserted. The tabulated ABR pure tone threshold data would further the argument about the patency of the hearing rescue in a compelling way.

Response: We thank the reviewer for the comment and valuable suggestions for improvement. We have provided a supplementary Table 1 of all ABR thresholds for experimental and control mice tested. Please see Supplementary Table 1. Also, we deleted the average waveform panel.

Frequency (kHz)	ABR Threshold (dB SPL), Atp2b2 ob1 ⁺ , Cas9: Atp2b2 -mut1 sgRNA, n=18																	
5.66	55	85	45	50	45	55	85	80	75	55	60	55	45	45	40	50	50	
8	45	75	40	35	60	60	70	75	65	40	50	45	35	40	50	50	50	
11.32	70	50	35	35	30	65	80	55	35	50	35	45	55	35	25	30	40	
16	80	35	35	35	35	40	70	45	45	40	35	65	65	35	60	40	70	
22.64	75	30	50	50	35	80	70	85	75	40	40	90	90		70	40	30	
32	85	50	55	90	45	85	85	85	60	50	80	90	85	70	65	60	90	
45.24	90	80	85	85	65	90	90	90	60	80	85	90	90	90	80	90	90	
Frequencies (kHz)	ABR Threshold (dB SPL), Atp2b2 ob1 ⁺ , uninjected, n=18																	
5.66	85	90	90	85	90	90	90	85	90	85	85	55	75	75	70	90	60	
8	80	90	80	70	70	90	90	90	70	90	75	75	70	75	80	90	80	
11.32		85	70	70	70	90	90	90	70	75	70	80	60	65	90	90	70	
16	90	75	75	65	85	80	90	75	75	50	55	90	90	65	80	70	70	
22.64	90	75	90	85	70	90	90	90	80	80	90	90	85	90	80	80	90	
32	90	90	90	90	90	90	90	90	90	90	90	90	90	90	90	90	90	
45.24	90	90	90	90	90	90	90	90	90	90	90	90	90	90	90	90	90	

L469: ". . . the edited hair cells may not be sufficient to maintain the health and function of the cochlea over time, leading to diminished cochlear function." Is the argument that unedited hair cells die and this structurally or electrically destabilizes the rescued hair cells, negatively impacting their function? Be

explicitly clear in your argument.

Response: We thank the reviewer for this comment. To maintain the whole cochlear function, 20% edited HC may not be sufficient to maintain the function of the cochlea that lacks 80% of functional HCs in the long term. The unedited HC will die, which may impact the structure as well as the function. We have added this hypothesis in the text. Please see p14, lines: 466-472.

Supplementary Figure 4a,b: Since the *Atp2b2*-mut 2 allele targets both wild type and mutant alleles, wouldn't elevated ABR and DPOAE thresholds in treated *Obl/+* animals be expected compared to controls? Why do you think this was not the case?

Response: We thank the reviewer for this comment. The *Atp2b2*-mut 2 sgRNA edits both WT and the mutant alleles. In the *Atp2b2^{obl/+}* mice, severe hearing loss was present at the time of the hearing test in uninjected ears. It is possible that further editing in the WT allele in some hair cells did not exacerbate pre-existed hearing loss. It may also be possible that editing of WT and mutant alleles happens in the same hair cells, which will lead to a hearing profile similar to uninjected ears.

Abstract L51: “. . .RNP delivery targeting both mutations lead to partial hearing rescue.” Leads to hearing rescue. Revise.

Response: We thank the reviewer for the comment. We have revised it to “leads”.

L88: “The auditory function requires functional inner and outer hair cells” Too many functions. Avoid the repetition of the same word.

Response: We thank the reviewer for the comment. We have revised it to “The auditory process requires functional inner and outer hair cells”.

L89: “. . . Further, multiple genetic hearing loss has been shown to be associated with mutations. . .” Clarify what is meant by multiple genetic hearing loss. Also, its associated with is better here. Revise.

Response: We thank the reviewer for the comment. We have revised this sentence. “Besides, multiple types of genetic hearing loss are associated with mutations in OHC.” Please see p3, lines: 85-86.

L92: “OHC function shown by distortion product of acoustic emissions” Provide the correct articulation of DPOAE here.

Response: We thank the reviewer to point it out. It should be “distortion product otoacoustic emission”.

L100: Provide Obl reference here.

Response: We thank the reviewer for the comment. We provided Obl reference here.

L135: consider protospacer rather than spacer.

Response: We thank the reviewer for the comment. We changed it to “protospacer”.

L137: Consider adding the cell line for the in vitro analyses here.

Response: We thank the reviewer for the comment. We added the cell line editing results here. Please see p4-5, lines: 135-150.

L909: Consider replacing the asterisks in 1b with the actual base pair sizes of the fragments. Same for Supplementary Figure 1a.

Response: We thank the reviewer for the comment. We revised Figure 1b as requested, as well as Supplementary Figure 1a.

Figure 2a: Genomic DNA rather than Genome DNA. Revise.

Response: We thank the reviewer for the comment. “Genome DNA” was changed to “Genomic DNA”.

L210: rate of 0.4~1.2%. Remove the approximately symbol and replace with a hyphen.

Response: We thank the reviewer for the comment. We revised it to “0.4-1.2%”.

L275: instead of “in a significant number of hair cells, why not say in an estimated 21% of hair cells.”

Response: We thank the reviewer for the comment. We revised it to “in an estimated 21% of hair cells”.

L343: State the frequency range over which the ABRs were conducted here.

Response: We thank the reviewer for the comment. We have stated the frequency range over which the ABRs were conducted here. Please see

“Uninjected *Atp2b2*^{Obi/+} inner ears showed elevations in cochlear neural responses at 4 weeks of age, with ABR thresholds ranging from 70-90 dB, compared with the 30-40 dB for wildtype C3H mice from 5.66 to 45.24 frequencies (Fig. 5b)”. Please see p10, lines: 332-335.

Figure 5C: Define the gray boxes in the figure.

Response: We thank the reviewer for the comment. It was meant to show the lowest thresholds detected. We have deleted the gray boxes as the red lines serve the purpose.

Reviewer #2 (Remarks to the Author):

The authors have performed several new experiments and the manuscript has improved.

I only have one comment for Figure 2e in the revised manuscript. On this Figure, the authors present new nested PCR analysis of the wild-type and mutant DNA, to investigate the presence of large deletions. The gel image is very inconsistent for the following reasons:

- 1) a different DNA ladder is used for the mutant allele and the wild-type allele.
- 2) for the WT, only one gel is shown, while for the mutant allele, two gels are shown that represent the two different PCRs.
- 3) In the uninjected mutant mice - the three animals show a different PCR result - the first animal has a smeary band, the second has 4 short bands and the last one has one band at around 100bp. It is not possible to conclude based on this PCR if the result from 3 different animals show 3 different profiles
- 4) The WT gel is cropped but short bands are half-visible. Are these primer dimers? If around 100bp, they are too long.

In general the PCR has to be optimised and the results should be presented side-by-side.

Response: We thank the reviewer for the comment. We redid the PCR of the injected and uninjected samples and ran a gel to show the results. We added 3% DMSO into the PCR reaction to reduce primer dimers. The result can be seen in the new Fig. 2e. The extra bands (red asterisks) were only detected in the injected *Atp2b2*^{Obi/+} ears and showed large deletions by NGS. The bands below the extra bands were spurious PCR products that yielded non-specific GNS reads. See p7, lines 220-223, and p27, lines 916-920.

e